# Topological scoring of protein interaction networks

Mihaela E. Sardiu[1], Joshua M. Gilmore[1,3], Brad D. Groppe[1,4], Arnob Dutta[1,5], Laurence Florens[1] & Michael P. Washburn [1,2]

It remains a significant challenge to define individual protein associations within networks where an individual protein can directly interact with other proteins and/or be part of large complexes, which contain functional modules. Here we demonstrate the topological scoring (TopS) algorithm for the analysis of quantitative proteomic datasets from affinity purifications. Data is analyzed in a parallel fashion where a prey protein is scored in an individual affinity purification by aggregating information from the entire dataset. Topological scores span a broad range of values indicating the enrichment of an individual protein in every bait protein purification. TopS is applied to interaction networks derived from human DNA repair proteins and yeast chromatin remodeling complexes. TopS highlights potential direct protein interactions and modules within complexes. TopS is a rapid method for the efficient and informative computational analysis of datasets, is complementary to existing analysis pipelines, and provides important insights into protein interaction networks.

[1] Stowers Institute for Medical Research, Kansas City, MO 64110, USA. [2] Department of Pathology and Laboratory Medicine, The University of Kansas Medical Center, 3901 Rainbow Boulevard, Kansas City, KS 66160, USA. [3] Present address: Boehringer Ingelheim Vetmedica, St. Joseph, MO 64506, USA. [4] Present address: Thermo Fisher Scientific, Waltham, MA 02451, USA. [5] Present address: Department of Cell and Molecular Biology, University of Rhode Island, 287 CBLS, 120 Flagg Road, Kingston, RI 02881, USA. Correspondence and requests for materials should be addressed to M.P.W. (email: mpw@stowers.org)

Many large- and medium-scale analyses of protein interaction networks exist for the study of protein complexes[1–4]. These studies typically consist of affinity purifications of different bait proteins analyzed using mass spectrometry (AP-MS) and utilize statistical tools to provide confidence that a prey protein is associated with a bait protein. Approaches like CompPASS[5], QSPEC[6], SAINT[7], and SFINX[8] largely yield statistical values, like a p value, to provide a confidence that two proteins are associating or are part of a protein complex. Within a protein interaction network, an individual protein may have multiple interactions, may be part of a large protein complex or complexes, which can be composed of important functional modules. For example, modularity is a hallmark of protein complexes involved in transcription and chromatin remodeling. Within this area of protein interaction networks, Mediator[9], SAGA[10,11], and SWI/SNF[12] are just a few of the many complexes well known to have modules that carry out distinct functions. Determining these modules in these complexes has required years of study using biochemical, genetic, and proteomic methods[9–12]. In addition, within protein interaction networks and protein complexes, there are also direct protein interactions that are critical for biological functions. For example, in DNA repair, the formation of the Ku70-Ku80 (XRCC5-XRCC6) heterodimer is critical for recognition of DNA double strand breaks that occur during nonhomologous end joining[13]. Existing statistical tools struggle to gain insight into the behavior of an individual protein in a protein interaction network. Methods are needed to dig deeper into protein interaction datasets to determine direct protein interactions and to capture modularity within complexes.

We have used approaches like deletion network analyses, network perturbation, and topological data analysis to determine the modularity in protein interaction networks and protein complexes themselves[14]. Here we describe a topological scoring (TopS) algorithm for the analysis of protein interaction networks derived from quantitative proteomic AP-MS datasets. TopS can be used by itself or in addition to existing tools[5–8] to analyze and interpret datasets. Here, we apply TopS to a protein interaction network centered on human DNA repair proteins, to a previously published human polycomb complexome dataset[4], and to yeast chromatin remodeling datasets from the INO80 and SWI/SNF complexes[12,14]. TopS yields insights into potential direct protein interactions and modularity within these networks. TopS is a simple and powerful algorithm based on the likelihood ratio method to infer the interaction preferences of proteins within a network consisting of reciprocal and nonreciprocal purifications. The TopS algorithm generates positive or negative values across a broad range for each prey/bait combination relative to the other AP-MS analyses in a dataset. TopS can differentiate between high-confidence interactions found with large positive values and lower confidence interactions found with negative values. TopS has the advantage that the scores it generates can be easily further integrated into additional computational workflows and clustering approaches.

## Results

### Definition of topological score based on likelihood ratio.
Determining whether a protein in a single affinity purification is significantly enriched without additional information remains a challenge. Therefore, by comparing several biologically related baits, one can determine whether a protein is truly enriched in a sample. The concept behind computing TopS is to collectively analyze parallel proteomics datasets and highlight enriched interactions in each bait relative to the other baits in a larger biological context. For each individual bait, instead of calculating a score by concentrating only on a single bait column via normalization or modeling, we aggregate information from the whole dataset where all data from all rows and all columns are used. Our topological score is based on the likelihood ratio and reflects the interaction preference of a prey protein for an affinity-purified bait. For each protein detected in a bait AP-MS, TopS calculates the likelihood ratio between the observed spectral count $Q_{ij}$ of a protein $i$ in a bait $j$, and the expected spectral count $E_{ij}$ in row $i$ and column $j$ (Fig. 1, Eq. (2)). Each prey protein from each AP-MS experiment has a distinct TopS value. TopS assigns positive or negative scores to proteins identified in each bait using spectral count information. If the actual number of spectra of a prey protein in a specific bait AP-MS exceeds that in all baits AP-MS, the presumption is that we have a positive preferential interaction. Likewise, fewer spectra in the bait AP-MS indicate a negative interaction preference.

Unlike p values or fold changes where the difference between the largest and the smallest values is relatively small, TopS generates a wide range of positive and negative scores that can easily differentiate high, medium, or low interaction preferences within the data. This is an advantage for proteomics data analysis since these scores not only reflect the interaction preference of proteins relative to others, but can now be directly integrated to further analyses such as clustering or network analysis in order to discover network organization. TopS is written in R and the platform is built with SHINY (https://shiny.rstudio.com/). It is easily implemented and includes correlations and clustering for bait/prey relationships.

### Analysis of a human DNA repair network dataset.
We first tested TopS on a dataset generated from HaloTag proteins involved in human DNA repair. DNA repair mechanisms are complex, independent, interdependent, and have been extensively studied[15,16]. To uncover the connectivity between proteins involved in these pathways, we selected 17 proteins that are part of different DNA repair mechanisms. In addition to the affinity purifications of known elements of the DNA repair pathways such as MSH2, MSH3, and MSH6 (involved in mismatch repair), RPA1, RPA2, and RPA3 (involved in nucleotide excision repair), XRCC5 and XRCC6 (involved in double strand break repair), SSBP1 and PARP1[17], we analyzed an additional seven proteins (WDR76, SPIN1, CBX1, CBX3, CBX5, CBX7, and CBX8) with chromatin associated functions and some of which had been associated with DNA repair[18]. For example, we have previously demonstrated that WDR76 is a DNA damage response protein with strong associations with members of DNA repair pathways and the CBX proteins[19]. It is important to note that our objective here was not to describe a human DNA repair protein interaction network. The proteins chosen for this small-scale study were of interest because of their potential relationship to the poorly characterized WDR76 protein[19]. Furthermore, these proteins were transiently overexpressed in HEK293 cells, and given the potential issues with transient overexpression[20], the dataset would be expected to be noisy. We reasoned that using a noisy dataset would be an excellent test of the TopS method and its ability to extract meaningful biological information.

We used Halo affinity purification followed by quantitative proteomics analysis to identify proteins associated with any of the 17 baits. Three biological replicates were performed for each of the bait proteins and the distributed normalized spectral abundance factor (dNSAF)[21] was used to quantify the prey proteins in each bait AP-MS. To eliminate potential nonspecific proteins, three negative controls were analyzed from cells expressing the Halo tag alone. A total of 54 purifications were completed and 4509 prey proteins identified (Supplementary

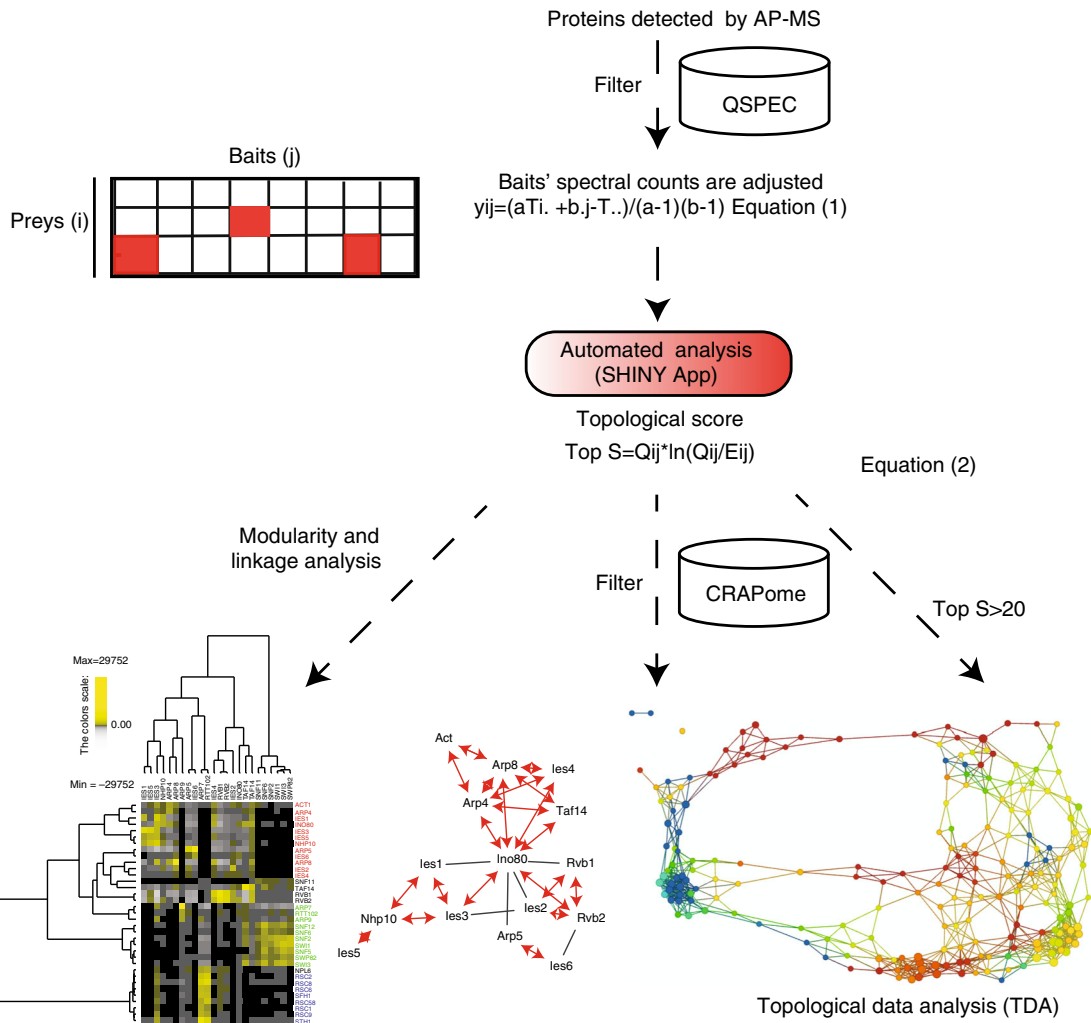

**Fig. 1** Workflow for topological scoring of protein interaction datasets. In the first step, contaminant proteins are filtered using $Z$ scores and FDR generated from QSPEC; however, other approaches could be used. Next, overexpressed bait proteins that have high spectral counts in the data are adjusted using Eq. (1). After prey proteins as baits are adjusted, topological scores (TopS) are then calculated, using an automated SHINY application, as described in Eq. (2), where $Q_{ij}$ is the observed count in row $i$ and column $j$ and $E_{ij}$ is the expected count in row $i$ and column $j$. Direct input of TopS values can be used in many different ways, such as data clustering to investigate the modularity and linkage in a network; additional filtering may be conducted using the CRAPome[27], for example; and a topological data analysis network may be generated. FDR false discovery rate

Data 1). To determine the proteins that were enriched in the samples versus negative controls, QSPEC[6] was used (Supplementary Data 1). A protein was considered specific in a sample if its $Z$ score was greater than or equal to 2 and the FDR was less than 0.01 in the bait AP-MS versus the control AP-MS. A total of 801 prey proteins passed these strict statistical criteria and they were further used in the analysis. Because of the higher number of spectra identified for overexpressed bait proteins, we adjusted the spectral counts for each bait protein according to Eq. (1) (Fig. 1). To depict the interactions in this DNA repair dataset that consisted of a matrix of 801 proteins in 17 baits, we calculated topological scores based on Eq. (2) and assigned positive or negative TopS to proteins identified in each bait AP-MS. The FDR values for these proteins using the QSPEC[6] pipeline were also calculated (Supplementary Data 1 and Supplementary Figure 1). To focus on proteins with positive interaction preferences, we used a TopS cutoff of 20 (Supplementary Figure 2). A total number of 617 proteins passed this filtering criteria (Supplementary Data 2).

Next, we hypothesized that proteins within the same complexes should have high preferences to the same bait purification. To test

this hypothesis, we sorted proteins into known complexes using ConsensusPathDB[22] and detected 118 protein complexes consisting of 230 proteins. Some of these 230 proteins were shared by multiple protein complexes. We systematically examined their TopS values and found that indeed proteins within complexes tended to associate with the same baits with high topological scores (Fig. 2 and Supplementary Data 3) even though some of the proteins were detected in most of the baits. To further investigate this observation, we hierarchically clustered all of the proteins belonging to known complexes and obtained a strong separation of the baits: bait proteins that belong to the same complexes clustered together (Fig. 2a). Thus, by using TopS values, we could illustrate the preferential interactions between protein complexes and baits in a large dataset.

Certain complexes showed significant enrichment to some of the baits. For example, in the case of the polycomb complex, we detected five members of the complex including CBX8. These five proteins had high preferential interactions to the CBX8 bait (Fig. 2b). Similar results were observed in the case of the BRAFT complex where we observed that five components of the complex, including RPA proteins, show high scores specifically

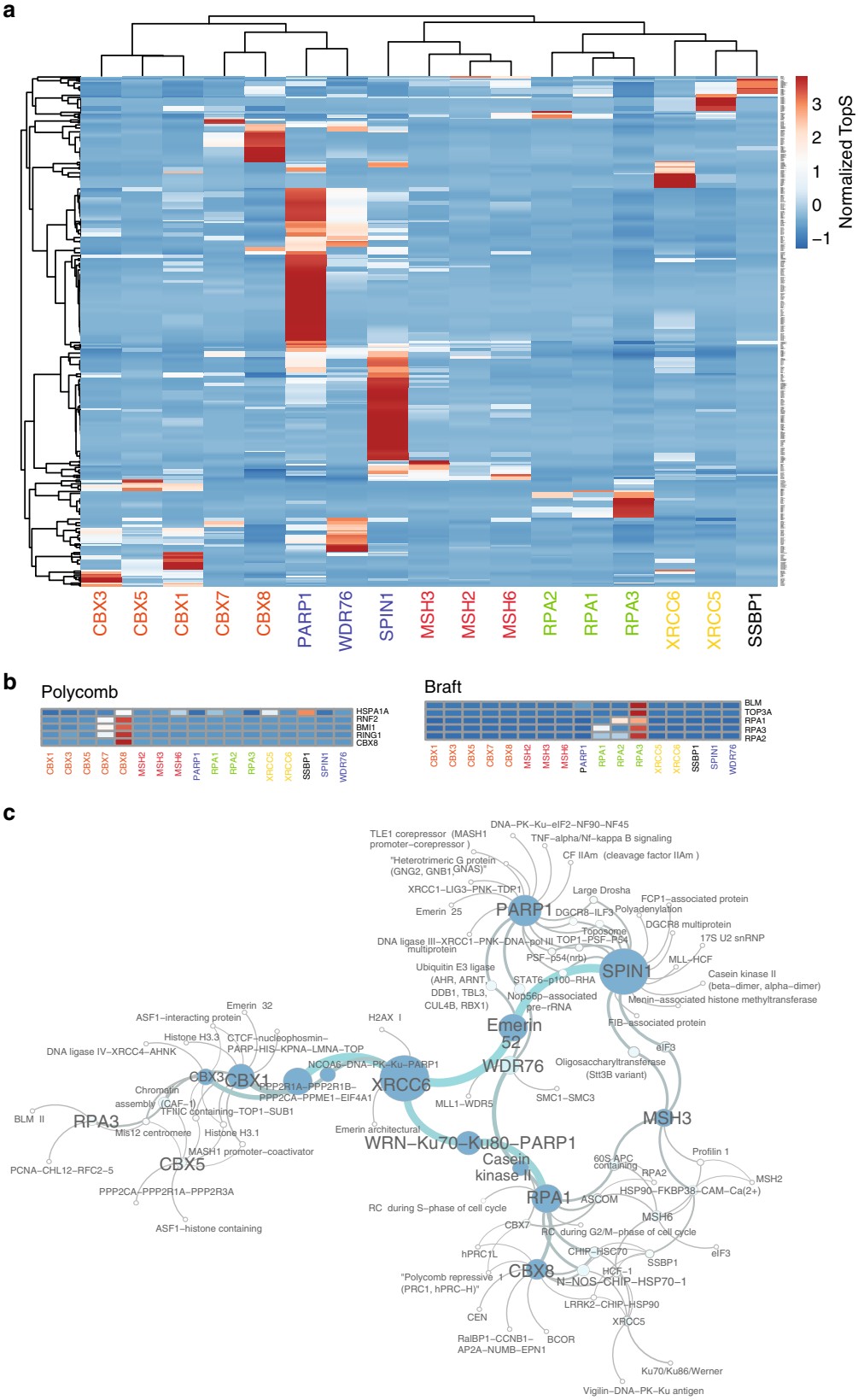

with the RPA3 bait AP-MS dataset confirming the connectivity between these proteins (Fig. 2b). The high TopS value observed with RPA3 suggests that this subunit of the RPA1/RPA2/RPA3 module brings in the other two BRAFT-specific proteins. Finally, to further illustrate all of the connections between complexes and baits, we constructed a network using the Cytoscape platform[23] (Fig. 2c). This network showed that PARP1, SPIN1, and XRCC6 baits have the most connections, and PARP1 and SPIN1 share a significant connected subnetwork (Fig. 2c).

**Fig. 2** Capture of protein complexes in a human DNA repair network. **a** Hierarchical biclustering on TopS values of the 17 baits and 535 preys in a human DNA repair protein interaction network. Proteins identified in complexes in the DNA repair dataset were hierarchically clustered using normalized TopS values as input using ClustVis[46]. **a** Scaling method was used to divide the values by standard deviation so that each row had a variance equal to one. Rows were centered and unit variance scaling was applied to rows. Rows were clustered using correlation distance and average linkage. Columns were clustered using correlation distance and Ward linkage. **b** Bait-specific protein complex enrichment. Two complexes are shown that exhibited high association with specific baits. Red color corresponds to high TopS scores. Subunits of the polycomb complex exhibited high scores with the CBX8 bait, and components of the BRAFT complex exhibited high scores with the RPA3 bait. **c** Interaction network between baits and known protein complexes identified in the dataset. The network was constructed using the Cytoscape platform[23]. Large nodes correspond to a larger number of links. TopS topological scoring

**Topological network assembly and extreme value analysis.** We have previously demonstrated that topological data analysis (TDA)[24] can be used to identify clusters of proteins in protein interaction networks using normalized correlations[25] and to find topological network modules in perturbed protein interaction networks using fold change ratios[14]. We then assembled a protein interaction network based on TDA using TopS values as input (Fig. 3a). TDA in conjunction with TopS values resulted in proteins with high topological scores clustering to the same region of the network (Supplementary Data 4). For example, proteins that are likely to interact with PARP1 were located in the same topological area on the right side of Fig. 3, whereas proteins that interact with WDR76 were on the upper left side. WDR76 localized in a node containing another 18 proteins including HELLS, GAN, and SIRT1, which have also been observed by independent studies to associate with WDR76[2,26]. Similarly, SPIN1 was in a node with another 18 proteins, four of which (SPIN1, SPIN4, THRAP3, and BCLAF1) have been reported by others in SPIN1 purifications[2].

Many additional approaches may be used to further interpret the dataset. For example, we introduced another filter to the TopS associations by adding information from the CRAPome, which is a database of proteins known to be present in 411 negative controls for human affinity purifications[27]. The list of proteins with high TopS values were analyzed using the CRAPome web interface. We next removed proteins with high TopS values in our dataset that were present in a maximum of 10/411 controls (~2% of the negative controls) with a maximum spectral count of 15. Using this threshold, we created a reduced network and inspected these specific interactions (Fig. 3b). For simplicity, we focused on the interactions of the CBX proteins where we observed a high connection between a set of zinc finger proteins with KRAB domains and the CBX proteins (Fig. 3b). Most of these zinc finger proteins were not identified in any of the 411 negative controls and are likely specific to the current dataset. KRAB zinc finger proteins play an important role in the evolution of gene regulatory networks[28]. While one of these KRAB proteins, TRIM28/KAP1, has been shown to directly interact with HP1 proteins (CBX1, CBX3, and CBX5)[29,30], all others are previously undescribed high-confidence interactors. The reduced subnetwork presented in Fig. 3b suggests a large number of additional KRAB Zinc finger proteins associate with CBX proteins in unexplored processes.

The known direct interactions of HP1 proteins (CBX1, 3, and 5) with TRIM28[29,30] suggested we look deeper into TopS values in our dataset. We sorted the TopS values for all the prey proteins for each bait protein (Supplementary Data 2) from highest to lowest to observe the extreme negative and positive values in the dataset. The 28 prey proteins with the highest TopS values across the 17 human DNA repair bait purifications are shown in Fig. 3c. TRIM28 is the highest scoring protein with all three CBX/HP1 proteins. In every other bait, except RPA3, TRIM28 is among the ten most negative TopS values and TRIM28 is the most negative scoring protein in the CBX8, MSH6, and WDR76 baits (Supplementary Data 2). This pattern of a prey having very high

TopS values in specific baits and extreme negative values elsewhere in the dataset occurred for several additional bait−prey interactions. These included XRCC5 and XRCC6, which directly interact with each other to form a heterodimer[13]. In the XRCC5 affinity purification, XRCC6 is the highest scoring prey protein (Supplementary Figure 3), which is the highest TopS value in the entire dataset, and in the XRCC6 affinity purification XRCC5 is the highest scoring prey protein (Fig. 3c and Supplementary Data 2). XRCC5 and XRCC6 are among the most negative TopS values in several other bait purifications including CBX5, CBX8, MSH2, MSH3, MSH6, RPA2, RPA3, SPIN1, and WDR76 (Fig. 3c and Supplementary Data 2). These results suggest that extremely positive TopS values may suggest direct protein−protein interactions.

An alternative approach to visualize positive and negative TopS scores is found in Supplementary Figure 4, where four subnetworks are shown in different bait affinity purifications. Subunits of the elF3 complex showed the same high positive scores in the MSH3 and SPIN1 affinity purifications and negative scores in the WDR76 and SSBP1 affinity purifications (Supplementary Figure 4a). Also shown are proteins found in the CBX/HP1 interactions, proteins of the CEN complex, and MSH2 interacting proteins (Supplementary Figure 4b–d). In these cases, very high TopS values were calculated in selected baits compared to others. The similarity of TopS scores for proteins in complexes in different affinity purifications further highlights the ability of TopS scores to rapidly capture meaningful information from a dataset. In addition, this illustrates a degree of similarity for proteins in complexes in terms of interactions suggesting coherence in TopS negative and positive sign assignments. Similar results can be observed for multiple complexes and groups of biologically related proteins (Supplementary Data 3 and Supplementary Figure 4).

As an example of how to further utilize prey proteins with high TopS values, we next sought to investigate the therapeutic aspect of these interactions since proteins involved in DNA repair pathways are known for their role in human diseases such as cancer. We used WebGestalt[31] to perform drug−gene association enrichment for the proteins with high TopS scores in our dataset. The enrichment analysis resulted in nine identified drug classes (Fig. 4a). Unexpectedly, we observed that the enriched classes consisted of proteins with high topological scores for the same bait (Supplementary Data 5). For example, the association with dactinomycin was mostly enriched with proteins with high positive TopS scores for PARP1, SPIN1, and WDR76 (Fig. 4b and Supplementary Data 5), whereas proteins associated with tobramycin had high scores for SPIN1 (Supplementary Data 5). Dactinomycin is approved for use in treatment of several cancers, such as Wilms's tumor[32]. These enrichment results indicate that TopS can highlight interactions that may be targeted by drugs in a dataset.

**TopS analysis of yeast chromatin remodeling complexes.** The fact that the highest TopS value in the human dataset was

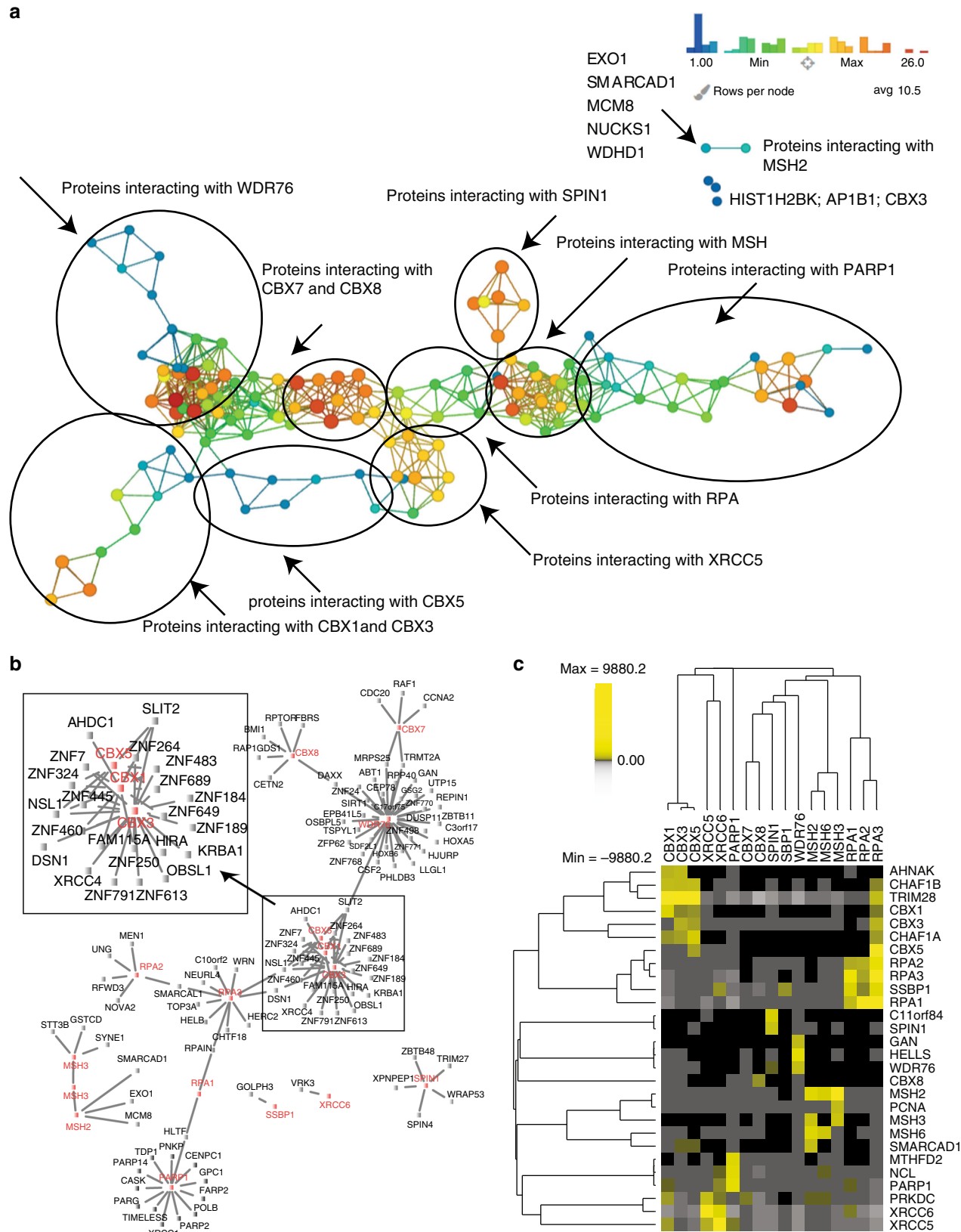

calculated for XRCC6 in the XRCC5 affinity purification, which is a known heterodimer[13], led us to investigate the TopS approach in the well-characterized yeast chromatin remodeling system. We have previously utilized deletion network analyses to determine the modularity of the INO80[14] and SWI/SNF[12] chromatin

remodeling complexes. Here, we reanalyzed two published yeast INO80[14] and SWI/SNF[12] protein complexes datasets for which crosslinking data also exist[33,34]. However, we only applied TopS to the wild-type affinity purifications and did not consider the affinity purifications in genetic deletion backgrounds. We sought

**Fig. 3** Topological network and linkage analysis of a human DNA repair network. **a** TDA in combination with the TopS scores was applied to the proteins with Tops > 20 in at least one of the baits. Norm Correlation was used as a distance metric with two filter functions: Neighborhood lens 1 and Neighborhood lens 2. Resolution 30 and gain 3 were used here using the Ayasdi software[24]. Proteins are colored based on the rows per node. Color bar: red: high values, blue: low values. Node size is proportional to the number of proteins in the node. **b** A reduced network was created using information from the CRAPome[27] database, and the network was generated using Cytoscape platform[23]. Proteins associated with CBX1, CBX3, and CBX5 are expanded and highlighted in the inset. **c** Hierarchical biclustering using PermutMatrix[47] of protein with extreme positive TopS values. The 28 proteins with the highest TopS values across the dataset are shown. Rows and columns were clustered using Pearson as the distance and Ward linkage as the method. Yellow corresponds to high TopS values and gray shows negative TopS values. Proteins not present in the purifications are shown in black. TopS topological scoring

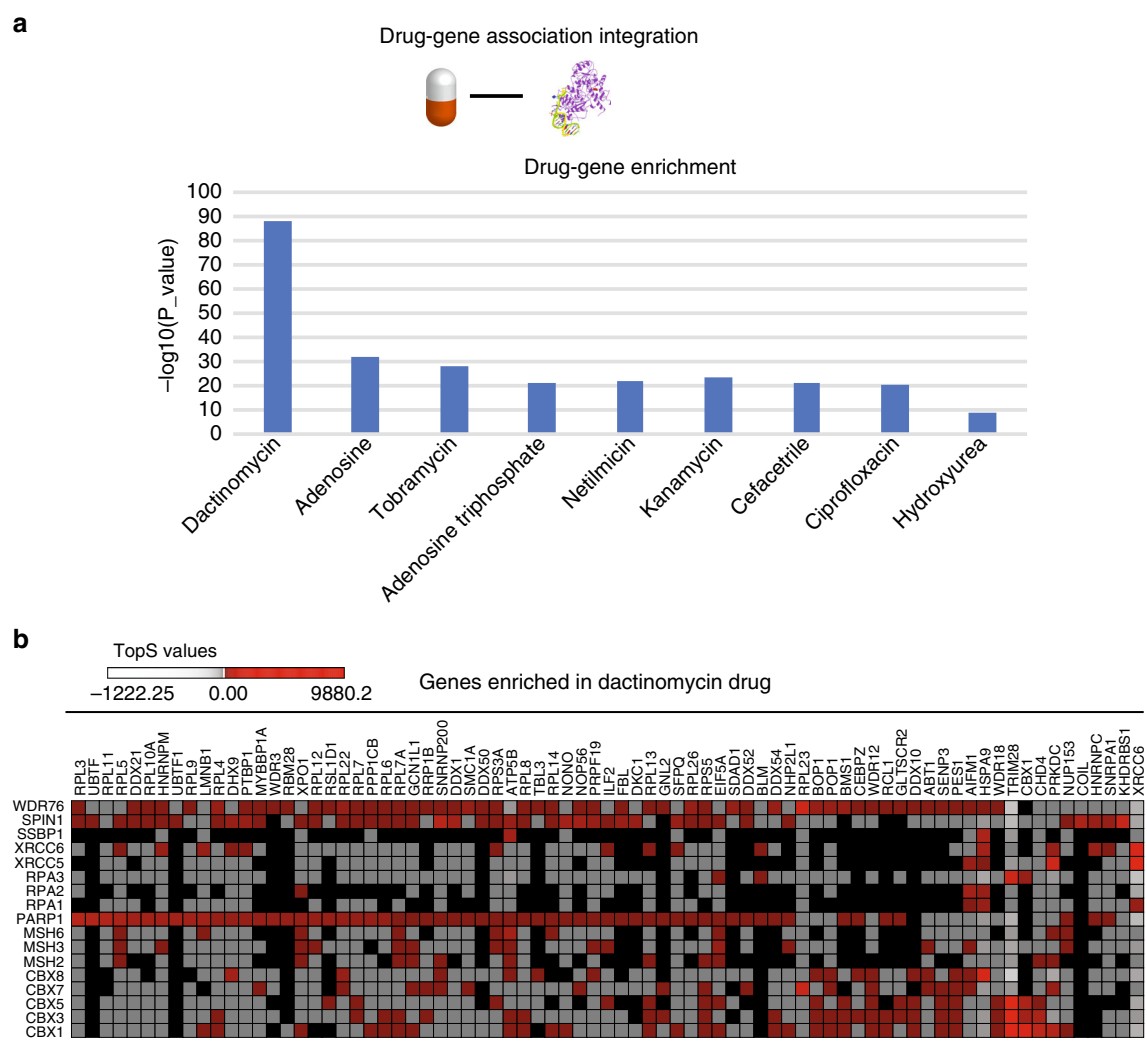

**Fig. 4** Enrichment of drug targets in the human DNA repair network. **a** The proteins detected and scored in our human DNA repair dataset were enriched in nine gene—drug classes. The enriched classes were obtained using WebGestalt database[31]. **b** Proteins/genes enriched in the dactinomycin set are represented. The red color represent high TopS values. TopS topological scoring

to determine if TopS analysis applied to the wild-type INO80 and SWI/SNF data could capture modules as determined from the deletion datasets and potential direct interactions as determined from crosslinking data.

Identifying direct interactions from quantitative wild-type affinity purification datasets has been a long-standing problem. Examining protein abundances in separate samples, one cannot identify the direct interactions using standard approaches. This challenge is illustrated in Fig. 5 where the protein abundances, as

estimated by dNSAF, of the INO80 complex in the ARP8 bait (Fig. 5a) and the protein abundance of the SWI/SNF complex in the SWP82 bait (Fig. 5b) are shown. For example, ACT1, ARP4, ARP8, IES4, and TAF14 are known to be part of a module based on deletion network analysis[14] and crosslinking data[34], yet their dNSAF abundances have no particular pattern.

To determine whether TopS values can capture the modularity and potentially direct protein interaction in these complexes, we merged a total of 24 wild-type affinity purifications for several

**a**

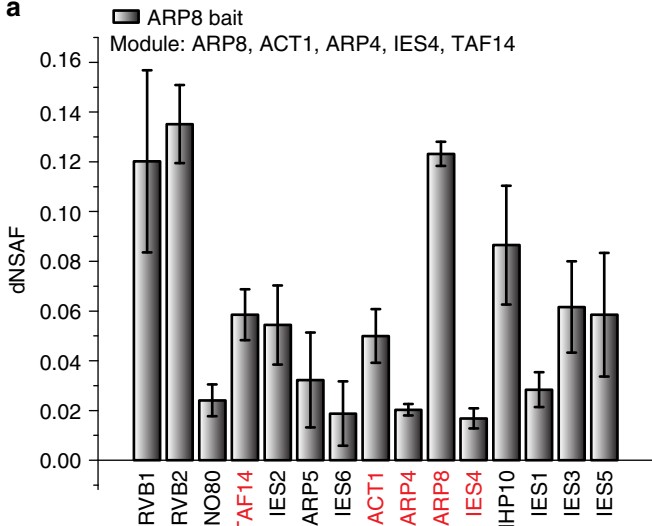

**b**

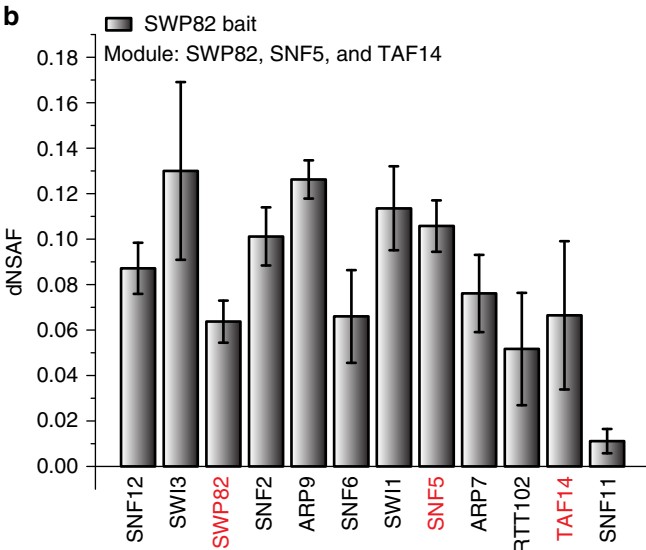

**Fig. 5** Failure to capture modularity in wild type-yeast complexes with a standard approach. Distributed normalized spectral abundance factors for the subunits of the INO80 and SWI/SNF complexes in APMS experiments using ARP8 and SWP82 as baits. Protein in red are included in **a** the ARP8, ACT1, ARP4, IEAS4, and TAF14 module, and **b** the SWP82, SNF5, and TAF14 module. Error bars represent one standard deviation of the dNSAF values from at least three biological replicates

INO80 and SWI/SNF subunits isolated from yeast (Supplementary Data 6 and 7). First, the INO80 data[14] were preprocessed by extracting nonspecific proteins, and proteins that passed the contaminant extractions in INO80 purifications (Supplementary Data 6) were also searched in the SWI/SNF dataset (Supplementary Data 7). A total of 237 proteins were shared between the two complexes. TopS analysis was applied to both complexes and FDR values for these proteins using QSPEC analysis were also obtained (Supplementary Data 9 and 10 and Supplementary Figure 5).

As with the human DNA repair network, we directly imported TopS values into a TDA-based network analysis (Fig. 6). In the INO80 complex, the ARP8 module was separated from the NHP10 and ARP5-IES6 modules (Fig. 6a, Supplementary Figure 6). Interestingly, the ARP8 module was connected via the shared protein TAF14 to a module consisting of SWI/SNF

subunits (Fig. 6a). In addition, the ARP9, ARP7, and RTT102 module in SWI/SNF was also clearly separated from the other modules (Fig. 6a, Supplementary Figure 7). ARP7, ARP9, and RTT102 are part of both the SWI/SNF[12] and RSC[35] complexes. Additional components of RSC were therefore present in the affinity purifications of these shared subunits and localized to an SWI/SNF subunit/RSC complex partition of the network (Fig. 6a). Overall, TopS values of wild-type affinity purifications were able to identify the modules of the INO80 and SWI/SNF complexes (Fig. 6a) in a manner comparable to the results from the deletion network analyses[12,14].

Next, similarly to how we processed the human DNA repair network, we sorted the TopS values for all the prey proteins for each bait protein (Supplementary Data 6 and 7) from highest to lowest to observe the extreme negative and positive values in the dataset. The 35 prey proteins with the highest TopS values across the 24 yeast chromatin remodeling wild-type bait purifications were hierarchically clustered (Fig. 6b). The highest TopS value in the entire dataset was ARP5 in the IES6 affinity purification. Furthermore, ARP5 and IES6 were the only two proteins in the IES6 affinity purification from the INO80 complex with positive TopS values; all the other proteins in the INO80 complex have negative values in the IES6 affinity purification (Fig. 6b and Supplementary Data 6 and 7). ARP5 and IES6 are well-characterized direct interactors in a subcomplex within the INO80 complex[14,36]. A similar result occurred in the ARP9 affinity purification where ARP7, RTT102, and ARP9 had the three highest TopS values, and all the other components of the SWI/SNF and RSC complexes had negative values in the ARP9 affinity purification (Fig. 6b and Supplementary Data 6 and 7). Again, ARP7, RTT102, and ARP9 are a known module since they are shared by both the SWI/SNF[12] and RSC[35] complexes.

An alternative approach to visualize positive and negative TopS scores in the INO80 and SWI/SNF datasets is found in Supplementary Figure 8, where six modules are shown in different bait affinity purifications. For example, in the INO8O dataset, the four proteins of the NHP10 module had high positive TopS scores in IES1 affinity purifications but negative values in the ARP5 and ARP4 affinity purifications (Supplementary Figure 8a). In the SWI/SNF dataset, the three proteins of the SNF5, SWP82, and TAF14 module had high TopS values in the SWP82, SNF6, and SNF2 baits, but negative values in the ARP7 bait (Supplementary Figure 8f). Similarly to the analysis of the human DNA repair dataset, modules in INO80 and SWI/SNF complexes showed similar patterns of interactions in different affinity purifications (Supplementary Figure 8), highlighting once again the accuracy of TopS positive and negative sign assignment to protein interactions.

We next evaluated the overlap of our high TopS values with reported crosslinking interactions from INO80[34] and SWI/SNF[33] (Fig. 6c, d and Supplementary Data 6 and 7). Our results showed a high overlap between crosslinking interactions and proteins −baits pairs with high TopS values. We observed 77% and 63% overlap for the INO80 and SWI/SNF complexes, respectively. The direct interactions identified by crosslinking are mostly between subunits located within the same module and TopS identified the majority of these interactions (Fig. 6c, d). The major exceptions were for proteins that are shared between different complexes like RVB1, RVB2, ARP7, ARP9, and RTT102 (Supplementary Data 6 and 7). In these cases, TopS give the highest scores to other complexes which share these proteins. For example, the module ARP7, ARP9, and RTT102 is shared with the RSC complex and we can see from Fig. 6b that members of the RSC complex are highly enriched in the ARP7 purifications. Overall, we found that TopS values from an analysis of wild-type INO80 and SWI/SNF AP-MS captured the known modularity of the complexes and

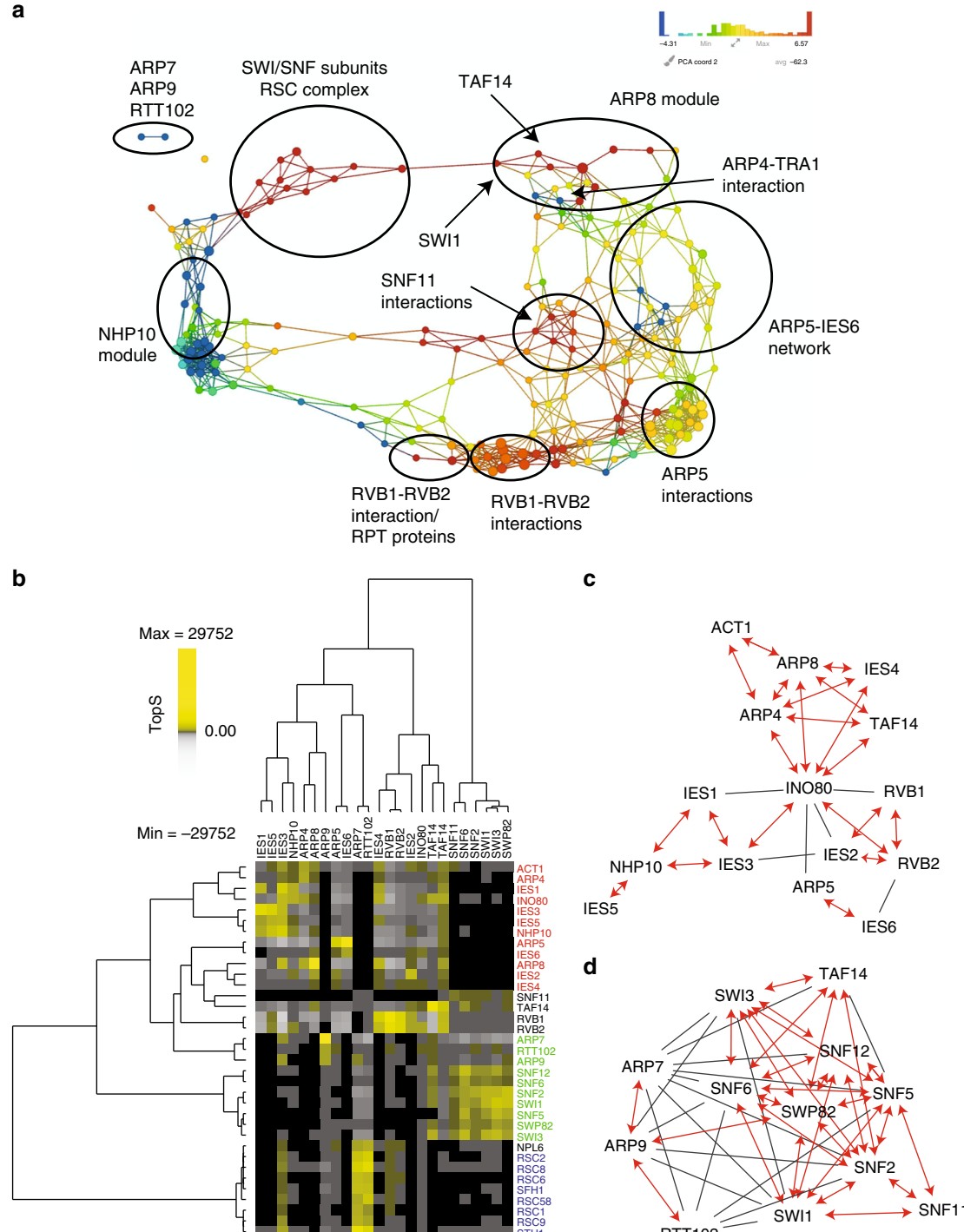

**Fig. 6** Topological network and linkage analysis of a yeast chromatin remodeling network. **a** A TDA network was constructed for the yeast data inputting TopS scores to the Ayasdi platform[24]. Distance Correlation was used as a metric with two filter functions: Neighborhood lens 1 and Neighborhood lens 2. Resolution 50 and gain 6x eq. were used. Proteins are colored based on the rows per node. Color bar: red: high values, blue: low values. Node size is proportional with the number of proteins in the node. The location of protein complexes subunits within the topological network are drawn on top of the TDA output. **b** Hierarchical biclustering using TopS values. Components of the INO80, SWI/SNF, and RSC complexes were clustered using PermutMatrix[47]. Rows and columns were clustered using Pearson as the distance and Ward linkage as the method. Yellow corresponds to high TopS values and gray indicates negative TopS values. Proteins that are not present in the purifications are shown in black. **c, d** Comparison of TopS values with crosslinking data. Each plot shows the direct interactions between subunits of **c** INO80 and **d** SWI/SNF. A red bidirectional edge represents the reported crosslinking interactions with a high TopS value. Black edges represent the reported crosslinking interactions with lower TopS values. TDA topological data analysis, TopS topological scoring

high TopS values correlated with known direct interactions within these protein complexes.

**Comparison of TopS to alternative analysis pipelines**. Several computational pipelines exist for the analysis of protein interactions generated from quantitative proteomic analyses of affinity purifications. These include the commonly used SAINT[7] and CompPASS approaches[5]. We applied these two pipelines to the human DNA repair and yeast INO80 and SWI/SNF datasets and compared the results to the TopS approach (Supplementary Methods, Supplementary Discussion, Supplementary Figures 9, 10, and Supplementary Data 8–10). First, as expected, we demonstrated that all three approaches effectively determine the components of protein complexes when bait purifications are compared to negative controls (Supplementary Figure 10a). Next, we analyzed the recall of potential direct protein interactions from all three approaches using human DNA repair proteins from BioGRID[37] and from the crosslinking-based results of the INO80 and SWI/SNF dataset. In this case, every affinity purification was compared to all other affinity purifications in the dataset, which were considered positive controls (Supplementary Methods, Supplementary Discussion). In all three cases, TopS showed improved recall over SAINT[7] and CompPASS[5] (Supplementary Discussion and Supplementary Figure 10b). As described earlier, an important feature of TopS is the large range of values it generates allowing the capture of modules and potentially the direct protein interactions in datasets.

One issue with such computational analyses is that the TopS algorithm may have been tailored to be particularly well-suited to the type of AP-MS datasets generated in our laboratory. To demonstrate the larger capabilities of the TopS algorithm, we reanalyzed an independently generated and previously published protein interaction network of the human polycomb complexome[4]. CompPASS[5] was also used to analyze this polycomb dataset that contains at least two biological replicates for 64 unique baits, for a total of 174 AP-MS containing 9853 candidate interactions[4]. As shown in Fig. 7a, TopS and TDA separated the interactions and revealed the cross-talk between different baits. For example, BAP1 and ASXL1/ ASXL2 baits were connected in a cluster on the left side of the topological area, with several interactions showing the same high scores to all three baits in agreement with the observations in the original study[4]. We also observed proteins that associated only with a single bait, such as SKP1 that linked to CSK interactions, while DSN1, ZN211, and TYY1 were isolated (Fig. 7a and Supplementary Data 11). Although the polycomb dataset contained spectral counts in an intermediate abundance range (Fig. 7b) when compared to our own human and yeast datasets described earlier, TopS recovered the majority of interactions (90% overlap with TopS ≥ 2 and WD$^N$ ≥ 1.5) reported in Hauri et al.[4].

As described earlier with the human DNA repair and yeast chromatin remodeling datasets, an important feature of the TopS approach is capturing extreme values. A biclustering analysis of selected baits and preys with high TopS values in the polycomb complexome dataset (Fig. 7c) revealed modules within the network including the BAP1, HCFC1, and OGT1 module seen in the ASX1, ASX2, OGT, and BAP1 baits (Fig. 7c). BAP1, HCFC1, and OGT1 are known interacting proteins[38–40]. Other modules captured in this analysis included a CBX2, PCGF4, RING1, PHC1, and PHC2 module, an EED, EZH2, SUZ12, MTF2, and JARID2 module, which are both known modules within the polycomb system[4], and a previously uncharacterized ADNP1, CHAP1, POGZ, and TIF1B module (Fig. 7c). Intriguingly, components of this last module, CHAP1 and POGZ, have been shown to interact and play a role in a rare form of

syndromic intellectual disability[41]. Overall, our TopS analysis of the polycomb complexome dataset[4] demonstrates the ability of the approach to rapidly analyze existing quantitative protein interaction network datasets, to generate distinct network visualizations, and to detect modules within these networks and potentially identify direct protein interactions.

## Discussion

To predict protein interactions from affinity purifications using quantitative proteomics, we have devised a TopS algorithm for evaluating the preference of each prey protein for a bait relative to other baits. We have built on the concept of TDA[24] as applied to protein interaction networks[14,25] to devise the TopS approach. Here, we have combined information from row, column, and total distributed spectral counts into this score to differentiate the preference of interaction with baits. This is specifically important for cases where proteins are detected in many runs. We have illustrated the methodology and its advantages through the analysis of two AP-MS datasets: a human DNA repair protein interaction network generated using transient transfections and a yeast chromatin remodeling protein interaction network. TopS values are directly incorporated into clustering and network assembly approaches and provide important insights into three protein interaction networks.

TopS values cover a broad and meaningful negative to positive range. In the human DNA repair proteins dataset, the highest TopS value was calculated for the XRCC6 prey in the XRCC5 affinity purification and these two proteins are known to interact and form a heterodimer[13]. Furthermore, XRCC5 and XRCC6 are among the most negative TopS values in several other bait purifications in the human DNA repair network dataset. The highest TopS value in the entire yeast chromatin remodeling dataset is ARP5 in the IES6 affinity purification, and again these two proteins are a well-characterized, directly interacting, submodule of the INO80 complex[14,36]. Again, ARP5 and IES6 had negative values in other INO80 bait purifications. Extreme positive TopS values in specific baits are typically reflected as extreme negative values in other bait purifications, even if these bait proteins are part of the same complex. This distinguishing feature of TopS provides important insights into proteins within a complex and suggests potential direct interactions. Furthermore, in the INO80 and SWI/SNF analysis, we were able to capture modularity from wild-type affinity purifications that we previously could only capture with the analysis of affinity purifications in deletion mutant backgrounds[12,14] and our results correlated strongly with crosslinking data[33,34]. A second important feature of TopS is its ability to identify modules within wild-type protein complex datasets.

The TopS platform has several important features. It is easy to implement. There are no parameters or assumptions of a probability distribution in our algorithm. The number of replicates can vary for the baits, where the replicates could be averaged by the user, without severely affecting the results. In addition, TopS is parameter free, distribution free, and independent of reference knowledge. TopS values can serve as a starting point or as a scaffold for other computational methods, visualization tools, and can be directly used in clustering and network analysis approaches. Lastly, multiple datasets can be integrated by merging their TopS scores. For straightforward usage by the community, TopS is implemented as a SHINY application (https://shiny.rstudio.com/). TopS is complementary to many computational pipelines used to process quantitative AP-MS datasets[5–8,42,43]. TopS can be used in addition to any of these approaches to provide a different perspective on a dataset. The TopS platform can be easily implemented in addition to these approaches to further analyze such datasets. If quantitative values are provided for all the prey

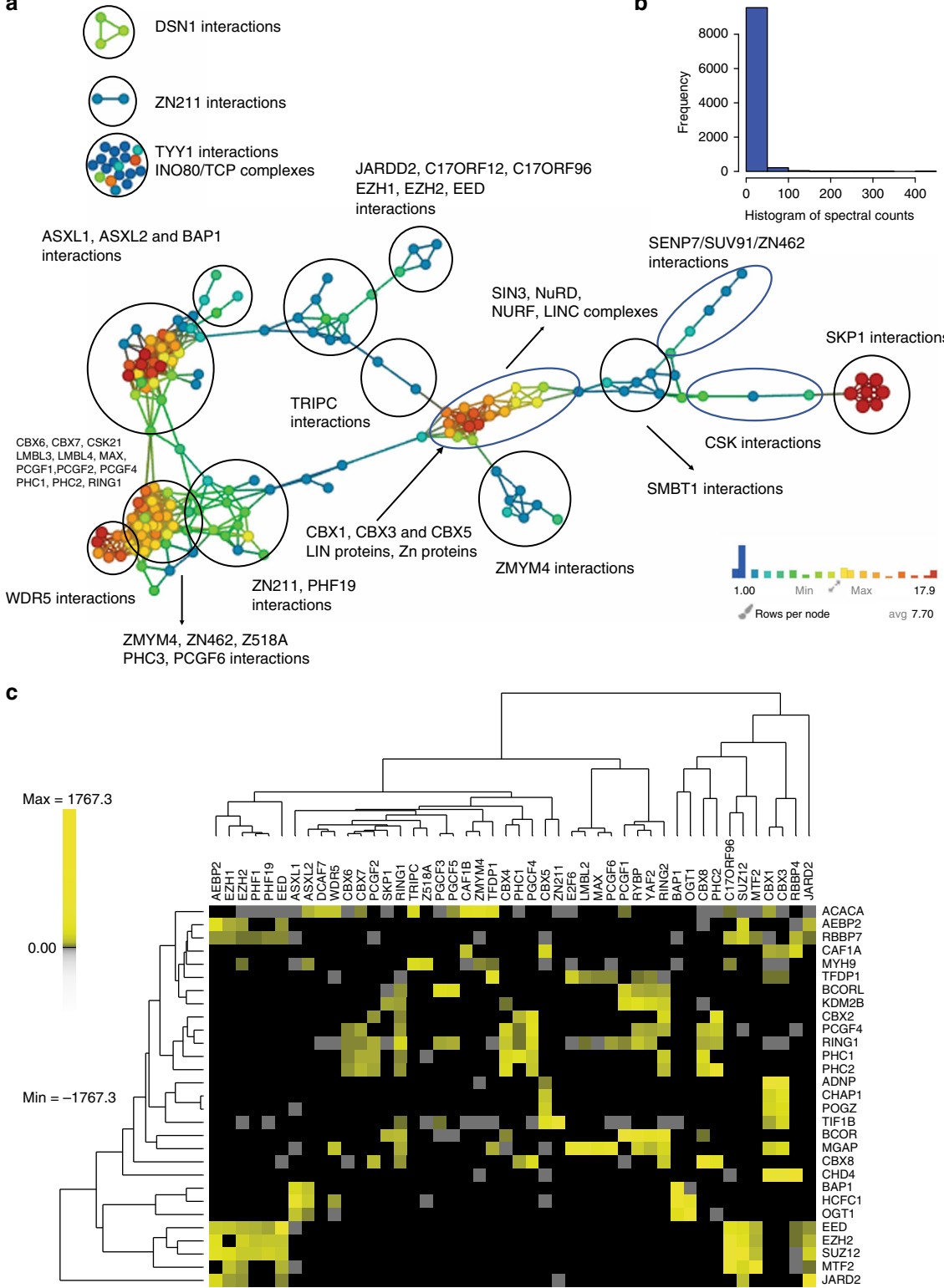

**Fig. 7** Topological network analysis of the polycomb complexome. **a** A TDA network was constructed using the Ayasdi[24] platform using TopS values as the input. Within the Ayasdi platform[24] Norm correlation was used for metric with two filter functions of Neighborhood lens 1 and Neighborhood lens 2, the resolution was set to 30 and a gain 4x eq. was used. Proteins are colored based on the rows per node. Color bar: red: high values, blue: low values. All proteins (408) that have a TopS value of 20 in at least one bait were included in here. **b** A histogram of the spectral counts for 9,853 candidate interactions was plotted. (**c**) Hierarchical biclustering on a subset of baits and preys within the dataset. Briefly, preys were included if they had a score of more than 100 in more than one bait and baits were excluded if none of the 32 proteins that passed this criteria had a score of 50 of higher in the bait. Also, RBBP4, RBBP7, CSK21, and CSK22 were removed since proteins passing the criteria above were only in RBBP4 and RBBP7 or CSK21 and CSK22. Rows and columns were clustered using Euclidean distance and Ward linkage as the method. Yellow corresponds to high TopS values and gray indicates negative scores. TDA topological data analysis, TopS topological score

proteins in a given bait, TopS can be used to reanalyze the data, like the recent description of the human polycomb complexome[4], for potential direct interactions and/or modules within protein complexes.

## Methods

**Materials**. Magne HaloTag magnetic affinity beads were purchased from Promega (Madison, WI). The following clones from the Kazusa DNA Research Institute (Kisarazu, Chiba, Japan) were used: Halo-WDR76 (FHC25370), Halo-XRCC5 (FHC07775), Halo-XRCC6 (FHC01518), Halo-RPA1 (FHC01462), Halo-RPA2 (FHC11655), Halo-RPA3 (FHC06678), Halo-MSH2 (FHC07773), Halo-MSH3 (FHC12698), Halo-MSH6 (FHC08173), Halo-CBX1 (FHC07438), Halo-CBX3 (FHC02188), Halo-CBX5 (FHC10519), Halo-CBX7 (FHC10535), Halo-CBX8 (FHC01705), Halo-PARP1 (FHC01012), Halo-SPIN1 (FHC10419), and Halo-SSBP1 (FHC07926). Flip-In HEK293T cells were purchased from Thermo Fisher Scientific (Waltham, MA) and authenticated using the ATCC (Manassas, VA) cell line authentication service. Cells are tested quarterly for mycoplasma and using the ATCC Mycoplasma detection kit (#30-1012K).

**Affinity purification and quantitative proteomic analysis**. Human proteins in the pFN21A plasmid with an N-terminal HaloTag were transiently transfected into HEK293T cells and whole cell extracts prepared and Halo affinity chromatography was performed on each independent whole cell lysate[44]. Briefly, 300 μl of whole cell extract diluted with 700 μl Tris Buffered Saline (TBS) was used for purifying Halo-tagged bait complexes using Magne HaloTag magnetic affinity beads (Promega, Madison, WI). The extracts were then incubated for 1 h at 4 °C with beads pre-pared from 100 μl bead slurry. The beads were then washed four times in buffer containing 50 mM Tris·HCl (pH 7.4), 137 mM NaCl, 2.7 mM KCl, and 0.05% NonidetP40. Bound proteins were eluted by incubating the beads for 2 h at 25 °C in 100 μl buffer containing 50 mM Tris·HCl (pH 8.0), 0.5 mM Ethylenediaminete-traaccetic acid (EDTA), 0.005 mM Dithiothreitol (DTT), and two units of AcTEV Protease (Invitrogen, Carlsbad, CA). To remove any traces of affinity resin, the eluates were spun through Micro Bio-Spin columns (BioRad, Hercules, CA). Samples were then processed and analyzed using label-free quantitative proteomic analyses using standard methods[19]. Briefly, after affinity purification of proteins using Halo affinity chromatography, samples were precipitated with trichloroacetic acid and centrifuged at 21,000 × g for 30 min at 4 °C. The resulting pellet was washed twice with acetone and resuspended in buffer containing 100 mM Tris·HCl (pH 8.5) and 8 M urea. The sample was treated with Tris(2-carboxyethyl)-phos-phine hydrochloride to reduce disulfide bonds, chloroacetamide (to prevent bond reformation), and digested with endoproteinase Lys-C for 6 h at 37 °C. After dilution to 2 M urea with buffer, samples were digested overnight with trypsin. Digested samples were then analyzed via multidimensional protein identification technology (MudPIT) on linear ion trap mass spectrometers (LTQ, Thermo Fisher Scientific, Waltham, MA)[19]. RAW files were converted to the ms2 format using RAWDistiller v. 1.0, an in-house developed software. The ms2 files were subjected to database searching using SEQUEST (version 27 (rev.9))[19]. Tandem mass spectra of proteins purified were compared against 29,375 nonredundant human protein sequences obtained from the National Center for Biotechnology (2012-08-27 release). Randomized versions of each nonredundant protein entry were included in the databases to estimate the false discovery rates (FDR)[19]. All SEQUEST searches were performed without peptide end requirements and with a static modification of +57 Da added to cysteine residues to account for carbox-amidomethylation, and dynamic searches of +16 Da for oxidized methionine. Spectra/peptide matches were filtered using DTASelect/CONTRAST[45]. In this dataset, spectrum/peptide matches only passed filtering if they were at least seven amino acids in length and fully tryptic. The DeltCn was required to be at least 0.08, with minimum XCorr values of 1.8 for singly, 2.0 for doubly, and 3.0 for triply charged spectra, and a maximum Sp rank of 10. Proteins that were subsets of others were removed using the parsimony option in DTASelect on the proteins detected after merging all runs. Proteins that were identified by the same set of peptides (including at least one peptide unique to such protein group to distinguish between isoforms) were grouped together, and one accession number was arbi-trarily considered as representative of each protein group. Quantitation was per-formed using label-free spectral counting. The number of spectra identified for each protein was used for calculating the distributed normalized spectral abun-dance factors (dNSAF)[21]. NSAF v7 (an in-house developed software) was used to create the final report on all nonredundant proteins detected across the different runs, estimate FDR, and calculate their respective distributed Normalized Spectral Abundance Factor (dNSAF) values.

**Topological scoring**. Since our approach aims to identify the enrichment of each protein in each bait relative to a collection of baits, overexpression of affinity-tagged bait proteins can diminish the interaction score. With this knowledge, we therefore selected to use a normalization method where the baits are estimated directly from the dataset.

To adjust for baits enrichment, we used this approach

$$y_{ij} = \frac{aT_{i.} + bT_{.j} - T_{..}}{(a-1)(b-1)}, \qquad (1)$$

where $a$ is the number of columns, $b$ is number of rows, $T_i$ represents the bait total spectral counts, $T_j$ is the row total spectral counts, and $T_{..}$ is the total spectral counts in the matrix. Using this approach, the estimated values of the bait proteins are now close to their average spectral counts in all the AP-MS runs of the dataset. Next, topological scores were calculated as follows. All the data that passed criteria from the QSPEC analysis was used as an input to the TopS determinations. We used a simple model to calculate a score for each prey–bait interaction as follows:

$$TopS = Q_{ij} \log \frac{Q_{ij}}{E_{ij}}, \qquad (2)$$

where $Q_{ij}$ is the observed spectral count in row $i$ and column $j$; and

$$E_{ij} = \frac{(\text{row sum } i)(\text{column sum } j)}{(\text{table sum})}. \qquad (3)$$

The data are treated in the following manner: (1) For each column and row, the sum of spectral counts is calculated; (2) the total number of spectral counts in the dataset is determined; (3) a TopS value for each protein in a bait AP-MS run is determined as described in Eq. (2). If the actual number of spectra of a prey protein in a bait AP-MS run exceeds that in all AP-MS results being analyzed, the presumption is that we have a positive interaction preference. Likewise, fewer spectra than in the AP-MS runs indicate a negative interaction preference. Proteins that were not detected in a particular run were assigned the value 0. A high positive score suggests a potential direct interaction between bait and prey. A negative score suggests that prey protein and bait interact to form a complex elsewhere in the dataset. We applied this framework to construct signed interaction networks derived from two independent AP-MS datasets obtained for yeast chromatin remodelers and human DNA repair proteins.

Next, TopS takes a numeric data matrix as input where multiple dimensions (e.g. proteins) are measured in multiple observations (e.g. baits/samples). In our case, the numerical values are distributed spectral counts. To make data input easier for the end user, we have defined the input file formats that include rows and columns annotation and numeric data. Any quantitative value, not only spectral counts, can therefore be utilized. TopS includes an example dataset for testing purposes: DNA repair dataset. TopS next generates an automatic output to make this type of analysis easier. MS Excel can be used to visualize the output and identify for example differences between samples or interactions between proteins and baits. Pearson correlation map, clustering analyses on initial numeric values (in our case distributed spectral counts), and TopS values are provided to the user as a pdf format output.

**Topological data analysis**. The input data for TDA are represented in a bait–prey matrix, with each column corresponding to purification of a bait protein and each row corresponding to a prey protein: values are TopS values for each protein. A network of nodes with edges between them is then created using the TDA approach based on Ayasdi platform (AYASDI Inc., Menlo Park, CA)[24]. Two types of parameters are needed to generate a topological analysis: First is a measurement of similarity, called metric, which measures the distance between two points in space (i.e. between rows in the data). Second are lenses, which are real valued functions on the data points. Lenses could come from statistics (mean, max, min), from geometry (centrality, curvature), and machine learning (PCA/SVD, Autoencoders, Isomap). In the next step the data are partitioned. Lenses are used to create overlapping bins in the dataset, where the bins are preimages under the lens of an interval. Overlapping families of intervals are used to create overlapping bins in the data. Metrics are used with lenses to construct the output. There are two parameters used in defining the bins. One is *resolution*, which determines the number of bins; higher resolution means more bins. The second is *gain*, which determines the degree of overlap of the intervals. Once the bins are constructed, we perform a clustering step on each bin, using single linkage clustering with a fixed heuristic for the choice of the scale parameter. This gives a family of clusters within the data, which may overlap, and we will construct a network with one node for each such cluster, and we connect two nodes if the corresponding clusters contain a data point in common. Norm Correlation was used as a distance metric with two filter functions: Neighborhood lens 1 and Neighborhood lens 2. Resolution 30 and gain 3 were used to generate Fig. 3a. and Resolution 50 and gain 6x eq. were used to generate Fig. 6a. Neighborhood lens 1 and Neighborhood lens 2 with the resolution 30 and a gain 4x eq. was used to generate Fig. 7a.

**Reporting summary**. Further information on experimental design is available in the Nature Research Reporting Summary linked to this article.

**Code availability**. TopS is written using Shiny application (R package version 3.4.2) for R statistics software. TopS uses several packages, including gplots, dev-tools and gridExtra. TopS is freely available at https://github.com/WashburnLab/Topological-score-TopS.

## Data availability

All the DNA repair data used here are deposited at the https://massive.ucsd.edu/ with MassIVE ID # MSV000081377. The data used for the analysis of the *S. cerevisiae* INO80 and SWI/SNF chromatin remodeling complexes are from previous studies on INO80 [14] (MSV000079138) and SWI/SNF (MSV000081417)[12]. Original data underlying this manuscript can be accessed from the Stowers Original Data Repository at http://www.stowers.org/research/publications/libpb-1280.

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

## Acknowledgements

Research reported in this publication was supported by the Stowers Institute for Medical Research and the National Institute of General Medical Sciences of the National Institutes of Health under Award Number RO1GM112639 to M.P.W. The content is solely the responsibility of the authors and does not necessarily represent the official views of the National Institutes of Health.

## Author contributions

M.E.S. and M.P.W. designed the experiments. J.M.G., B.D.G. and A.D. performed the experiments. M.E.S. performed computational analyses of data. M.E.S., L.F. and M.P.W. wrote the manuscript.

## Additional information

**Competing interests:** The authors declare no competing interests.

