## [Peer Review File · Nature Communications]

Reviewers' comments:

Reviewer #1 (Remarks to the Author):

The authors describe a new approach to scoring the results from affinity-purification mass-spectrometry (AP-MS) experiments called the Topological Score. They apply this core to new data from AP-MS experiments of proteins involved in DNA repair and to a previously described AP-MS study of chromatin remodeling complexes. They analyze these data using the TopS score in conjunction with visualization using the commercially available TDA package and show that they can identify biologically-relevant protein complexes and protein networks, and that the TopS score is able to rank prey proteins such that directly interacting proteins are ranked highest. As such, this new approach adds to the existing methodology available for analysis of AP-MS experiments. It is also intuitive and should be straightforward to implement / or to use the provided application (not tried by this reviewer).

Points for manuscript improvement:

1. The new TopS score would benefit from a side-by-side benchmark against one or more of the existing methods (saint, compass etc). This would enable better understanding of the advantages (or disadvantages) of the new scoring metric. With this kind of analysis, researchers using AP-MS would be better placed to select appropriate computational methodology for their own datasets. Precision-recall type analysis of how well the new metric can identify known components of protein complexes would improve the rigor of the study.
2. The TopS score explicitly takes into account the distribution of scores for each protein across all baits. the authors should explain in more detail how this differs from other methods. For example, computing a Z-score across prey proteins for all baits might be expected to yield similar information (assuming that there are sufficient instances of a given prey protein to analyze the distribution of scores).
3. The authors contrast the TopS metric with p-value based metrics for AP-MS data. One advantage to computing a p-value is that it has a specific meaning. The authors should explain more precisely how the TopS score should be interpreted. The authors state that the TopS metric advantage is that it is a broad range of values (from negative to positive). One advantage of this might be in how biologists could use this relatively easily-interpretable score.
4. The authors should explain more clearly why this is a topological score. In the strictest sense, this term applies to how objects are arranged in space - and so clearly applies to their chosen method of visualizing the protein networks. It's not so clear how it applies to the TopS metric, which considers distributions of how prey proteins occur across whole datasets of AP-MS experiments.
5. Explain how ad hoc thresholds are selected for filtering the data. For example proteins in less than 10/411 controls. This is a long standing issue in this area - and the authors could justify how these choices are made.

Comments on the manuscript:

Overall, this is a nicely presented study. It's clear and figures are nicely representative.

Check introduction - some sentences do not make sense e.g. "Existing statistical tools ..."
"TopS yield insights .."

Reviewer #2 (Remarks to the Author):

The manuscript by Sardiù describes a “topological score” (TopS) for improved analysis of APMS protein interaction data. The main idea behind this simple score is that a prey’s spectral count in a particular APMS run is converted to a TopS score by comparing it to spectral counts of all other preys for the same bait, and to spectral counts of that same prey across all other APMS runs. The high positive TopS score indicates strong (perhaps direct) interaction, whereas negative TopS score indicates indirect (or perhaps non-specific) interaction. The authors demonstrate that meaningful results can be obtained by clustering and visualizing (using hierarchical or topologic clustering) APMS data using TopS score as input.

While the method seems to work well, it is hard to distinguish it from many other methods for APMS data. In some way it is similar to that from the Gygi/Harper lab (see e.g. recent Bioplex and Bioplex 2.0 studies) - BioPlex method is based on the topological analysis as well (there is D score and entropy score). A similar normalization actually can be done as option in some of the hierarchical clustering algorithms. ProHits-Viz online tool computes what is called specificity scores. There is a MUSE score (<http://www.mcponline.org/content/15/9/3030.long>). There is also a MiST score (Krogan lab) that looks at both specificity and reproducibility (discussed in <https://www.ncbi.nlm.nih.gov/pmc/articles/PMC4332878/>)

There is a lot more literature on large interaction network analysis, where network topology has always been used. For example, the so-called socio-affinity score uses log-ratios of co-occurrences relative to what would be expected based upon protein purification frequencies.

I have a feeling that the authors find that their simple method performs really well, at least on data from their own studies, but they need to explain the novelty better. They also need to investigate the advantages and limitations of their method compared to other topology based methods on more datasets (different number of baits, different connectivity – e.g. connected vs disjoint networks).

Response to Reviewers' Comments

We would like to thank both reviewers for their supportive, insightful, and constructive review of our manuscript entitled “Topological Scoring of Protein Interaction Networks”. We have addressed all of the reviewers’ comments here in this response letter and in a significantly revised version of the manuscript. The major change includes a detailed comparison of the TopS algorithm to SAINT¹ and ComPASS² using the human DNA repair and yeast chromatin remodeling complexes described in the original version of the manuscript. In addition, to address the possibility that the TopS algorithm may have been tailored to be particularly well-suited to the type of APMS datasets generated in our laboratory we re-analyzed an independently generated and previously published protein interaction network of the human polycomb complexome that was analyzed using ComPASS³. This reanalysis is found in the new Figure 7. These comparisons are detailed on pages 15-17 in the main body of the revised manuscript and pages 2-5 of the revised supplementary information file. In addition, the revised manuscript now contains four additional supplementary tables (Supplementary Tables 8-11), and seven additional supplementary figures (Supplementary Figures 1, 4, 5, 7, 8, 9, and 10). In particular, Supplementary Figure 10 summarizes the comparison of the TopS algorithm to SAINT¹ and ComPASS² by comparing the recall of the three approaches and highlighting the dramatic differences in score distributions. Given the text length of no more than 5000 word for *Nature Communications* articles, the majority of the details regarding the comparison of pipelines is contained in the Supplementary Information file. As a result of the reviewers’ excellent critiques, we present a significantly improved body of work

Below we detail our response to each reviewer’s valuable and constructive critiques.

Response to Comments of Reviewer #1:

The authors describe a new approach to scoring the results from affinity-purification mass-spectrometry (AP-MS) experiments called the Topological Score. They apply this core to new data from AP-MS experiments of proteins involved in DNA repair and to a previously described AP-MS study of chromatin remodeling complexes. They analyze these data using the TopS score in conjunction with visualization using the commercially available TDA package and show that they can identify biologically-relevant protein complexes and protein networks, and that the TopS score is able to rank prey proteins such that directly interacting proteins are ranked highest. As such, this new approach adds to the existing methodology available for analysis of AP-MS experiments. It is also intuitive and should be straightforward to implement / or to use the provided application (not tried by this reviewer).

Response:

We would like to thank the reviewer for careful reading of our manuscript and for providing supportive and constructive comments.

Points for manuscript improvement:

1. The new TopS score would benefit from a side-by-side benchmark against one or more of the existing methods (saint, compass etc). This would enable better understanding of the advantages (or disadvantages) of the new scoring metric. With this kind of analysis, researchers using AP-

MS would be better placed to select appropriate computational methodology for their own datasets. Precision-recall type analysis of how well the new metric can identify known components of protein complexes would improve the rigor of the study.

Response:

This was an excellent and insightful recommendation made by both reviewers. As recommended by Reviewer #1 we compared TopS to both SAINT¹ and ComPASS² in the revised manuscript. In a section in the main body of the manuscript on pages 15-17 entitled “*Comparison of TopS to Alternative Analysis Pipelines*” we described these comparisons. In addition to recommending a comparative analysis, Reviewer #2 also questioned whether the TopS approach only worked on datasets generated in our laboratory. As a result of this question, the section entitled “*Comparison of TopS to Alternative Analysis Pipelines*” contains multiple comparisons of pipelines.

We first analyzed the human DNA repair and yeast chromatin remodeling complexes described in the original version of the manuscript using both SAINT¹ and ComPASS². The results of this comparison are summarized on page 15 of the manuscript and described in detail in the supplementary information document. Supplementary Figure 10 details the improved precision and recall of TopS over these approaches on 75 yeast interactions and 121 human interactions (see supplementary information file). Supplementary Figure 10 also highlights the differences in score distribution of TopS compared to these approaches. As described throughout the manuscript this is a key feature of TopS important for detecting modularity within complexes and datasets, for example.

Next, to address the broader applicability question regarding TopS we reanalyzed the polycomb complexome dataset analyzed using ComPASS from Hauri *et al.*⁴. The details of this re-analysis are described on pages 15-17 of the main body of the manuscript and visualized in the new Figure 7 of the revised manuscript. Figure 7 presents a new and informative analysis of the polycomb complexome with both a TopS based topological network in Figure 7A and a biclustering analysis of large TopS value bait and prey relationships in Figure 7B. Figure 7B highlights the power of the TopS approach to potentially detect direct interactions and capture modularity. One particularly interesting module not previously described by Hauri *et al.*⁴ is an uncharacterized module containing ADNP1, CHAP1, POGZ, and TIF1B where CHAP1 and POGZ, have been shown to interact and play a role in a rare form of syndromic intellectual disability⁵.

This reanalysis and these representations of the dataset are distinct from the network analysis and representations presented in Hauri *et al.*⁴ and demonstrates the ability of TopS to be applied to independently generated datasets. In addition, this reanalysis demonstrates that TopS can be used as a standalone method, but it is likely more powerful when used in concert with alternative methods.

2. The TopS score explicitly takes into account the distribution of scores for each protein across all baits. the authors should explain in more detail how this differs from other methods. For example, computing a Z-score across prey proteins for all baits might be expected to yield

similar information (assuming that there are sufficient instances of a given prey protein to analyze the distribution of scores).

Response:

Reviewer #1 is correct. If there are sufficient data points in the dataset (i.e. a larger dataset) and the dataset is normally distributed, one could apply Z-statistics. However, typical AP-MS runs are sparse where prey proteins are not detected in many baits the resulting matrix of data is not normally distributed. As a result we believe that the Z-score is not as well suited for the analysis of small or large AP-MS datasets. Also, TopS importantly uses a combination of row, column and total spectral count information for each protein topology while Z-score uses only the information in the row.

3. The authors contrast the TopS metric with p-value based metrics for AP-MS data. One advantage to computing a p-value is that it has a specific meaning. The authors should explain more precisely how the TopS score should be interpreted. The authors state that the TopS metric advantage is that it is a broad range of values (from negative to positive). One advantage of this might be in how biologists could use this relatively easily-interpretable score.

Response:

Reviewer #1 is correct that p-value based metrics are indeed valuable. In the revised manuscript we have now contrasted TopS with the FDR obtained from the QSPEC⁶. These comparisons are provided in Supplementary Figures 1 and 5, and the results provided in Supplementary Tables 8, 9 and 10, the results from TopS, QSPEC and FDR distributions. Sentences added on pages 7 and 12 highlight these additions. This information is provided but not discussed in great detail in the revised manuscript. Instead we focus on highlighting the unique features of TopS values that can be used in addition to p-values, for example.

We are pleased that Reviewer #1 finds the TopS values to be relatively easy to interpret and we fully agree with this assessment. In order to better describe the interpretation of TopS values we have added sections to the manuscript on pages 10 and 14 which describe an alternative ways of visualizing and using TopS values that are provided in supplementary figures 4 and 8. These two brief sections and two supplementary figures describe analyses of the human DNA repair and yeast chromatin remodeling datasets that further demonstrate another approach to visualize TopS values of proteins, modules, and complexes to gain new insights into protein interaction networks. That being said, we are quite interested to see how other researchers implement the TopS approach and we expect other researchers to devise other interesting interpretations of these values in the future.

4. The authors should explain more clearly why this is a topological score. In the strictest sense, this term applies to how objects are arranged in space - and so clearly applies to their chosen method of visualizing the protein networks. It's not so clear how it applies to the TopS metric, which considers distributions of how prey proteins occur across whole datasets of AP-MS experiments.

Response:

One of the challenges with the words ‘topology’ or ‘topological’ is that they are used widely in the scientific literature in many different fields. Here, our work is inspired by topological data analysis (TDA) which is a form of algebra ⁷. Our studies using TDA ^{8,9} led us to devise the TopS approach as an improved data entry into TDA itself. Briefly, topology can be used to abstract the inherent connectivity of objects while ignoring their detailed form. In mathematics, a metric or a distance function is a function that defines a distance between a pair of elements of a set. A set with a metric is called a metric space and it induces a topology on a set. An AP-MS data set matrix can be viewed as a network composed of multiple nodes connected by edges in a topological representation. The topological score is a measure between two nodes, like an edge, in which the nodes correspond to bait and prey proteins. The quantitative values in the datasets analysed in this manuscript provide the ‘height’ or ‘weight’ between two nodes. We also called this measure a topological score since TopS values are used to represent the data in three-dimensional space using for example the TDA approach.

5. Explain how ad hoc thresholds are selected for filtering the data. For example proteins in less than 10/411 controls. This is a long standing issue in this area - and the authors could justify how these choices are made.

Response:

The reviewer is correct that the filtering conducted in order to generate Figure 3B is indeed *ad hoc*. We selected this threshold of proteins present in 2% of the negative controls of the CRAPome ¹⁰ as a method to reduce the dataset to in order to highlight proteins enriched in the human DNA repair dataset. A protein present in more than 10 of 411 negative controls would likely be a contaminant. This was done as an exercise to demonstrate alternative approaches to interpret the dataset as described on page 9 of the manuscript. The purpose of using the CRAPome repository ¹⁰ was to highlight interactions which are specific to the DNA repair dataset and more likely true interactions. We choose a small threshold for the number of controls which corresponds to ~2% of the total number of controls. Reviewer #1 is correct that this is an issue in the area of protein interaction network analysis, but here this threshold was used only to reduce the network size for Fig. 3B and it does not change any of our results or conclusions.

Comments on the manuscript:

Overall, this is a nicely presented study. It's clear and figures are nicely representative.

Response:

We greatly appreciate Reviewer #1’s enthusiasm for our work and for their excellent recommendations to improve our manuscript.

Check introduction - some sentences do not make sense e.g. "Existing statistical tools ..." "TopS yield insights .."

Response:

We have carefully checked the manuscript and revised the necessary sentences for clarity.

Response to Comments of Reviewer #2:

The manuscript by Sardiù describes a “topological score” (TopS) for improved analysis of APMS protein interaction data. The main idea behind this simple score is that a prey’s spectral count in a particular APMS run is converted to a TopS score by comparing it to spectral counts of all other preys for the same bait, and to spectral counts of that same prey across all other APMS runs. The high positive TopS score indicates strong (perhaps direct) interaction, whereas negative TopS score indicates indirect (or perhaps non-specific) interaction. The authors demonstrate that meaningful results can be obtained by clustering and visualizing (using hierarchical or topologic clustering) APMS data using TopS score as input.

Response:

We wish to thank Reviewer #2 for their excellent summary of our manuscript finding that the TopS approach yields ‘meaningful results’. It was rewarding to read that both reviewers accurately summarized our research presented in this manuscript. We endeavoured in our research presented herein to develop a meaningful, interpretable, and implementable approach that would be of value to a wide range of researchers.

While the method seems to work well, it is hard to distinguish it from many other methods for APMS data. In some way it is similar to that from the Gygi/Harper lab (see e.g. recent Bioplex and Bioplex 2.0 studies) - BioPlex method is based on the topological analysis as well (there is D score and entropy score). A similar normalization actually can be done as option in some of the hierarchical clustering algorithms. ProHits-Viz online tool computes what is called specificity scores. There is a MUSE score (<http://www.mcponline.org/content/15/9/3030.long>). There is also a MiST score (Krogan lab) that looks at both specificity and reproducibility (discussed in <https://www.ncbi.nlm.nih.gov/pmc/articles/PMC4332878/>)

Response:

This was a very valuable recommendation, and as described earlier in the response to Reviewer #1’s first critique, both reviewers highlighted the importance to compare and contrast the TopS algorithm to existing approaches. Since it is not feasible to compare and contrast TopS to all existing approaches we focused our ‘**Comparison of TopS to Alternative Analysis Pipelines**’ on pages 15-17 of the revised manuscript to SAINT¹ and ComPASS², as suggested by Reviewer #1. Also, these two pipelines are arguably currently the most widely used since SAINT is the approach used in ProHits-Vis¹¹ and ComPASS is the approach used in Bioplex¹². As described previously in the response to Reviewer #1’s comment, the majority of the comparison of the approaches is presented in the Supplementary Information document and summarized in supplementary figure 10 where TopS had improved recall and precision over both approaches and a substantially wider scoring range than both approaches. In addition, from our re-analysis of the human polycomb complexome³, TopS can be used either as a standalone approach or in addition to existing approaches. However, it is likely that TopS is best implemented in addition to existing approaches, as we highlight at the end of the discussion on pages 18-19.

There is a lot more literature on large interaction network analysis, where network topology has always been used. For example, the so-called socio-affinity score uses log-ratios of co-occurrences relative to what would be expected based upon protein purification frequencies.

Response:

Reviewer #2 is correct that there is existing literature where network topology has been described. The socio affinity index, for example, was described in 2006 in an analysis of a yeast protein interaction network¹³. The socio affinity index was used in that case to study a large network with extensive numbers of reciprocal purifications¹³. It is not broadly applicable for this reason and is not suitable for analysis of smaller scale datasets. While valuable, covering the complete body of literature on network analysis and topology is not the intention of this manuscript. However, as described earlier, the word ‘topology’ is widely used in many fields and here we are refereeing to developing new approaches inspired by topological data analysis, which is a form of algebra.

I have a feeling that the authors find that their simple method performs really well, at least on data from their own studies, but they need to explain the novelty better. They also need to investigate the advantages and limitations of their method compared to other topology based methods on more datasets (different number of baits, different connectivity – e.g. connected vs disjoint networks).

Response:

Based on this important comment from Reviewer #2, we carried out a reanalysis of an existing and independently generated analysis using ComPASS of the polycomb protein interaction network³ as described earlier and on pages 15-17 of the revised manuscript in the section ‘*Comparison of TopS to Alternative Analysis Pipelines*’. As we detail in this section of the manuscript and in Figure 7, the TopS approach provides new insights into this network by providing a TopS based TDA network which shows linkages between portions of the network and by using TopS values in a biclustering approach to identify modules in this network. As described in the response to comment #1 from Reviewer #1, Figure 7B highlights the power of the TopS approach to potentially detect direct interactions and capture modularity. One particularly interesting module not previously described by Hauri *et al.*⁴ is an uncharacterized module containing ADNP1, CHAP1, POGZ, and TIF1B where CHAP1 and POGZ, have been shown to interact and play a role in a rare form of syndromic intellectual disability⁵. The ability to detect direct protein interactions and modularity in the datasets analyzed in the manuscript is a key and novel feature of TopS. The concern that it only works on data generated in our laboratory is valid and was important for us to investigate. Our reanalysis of the polycomb protein interaction network generated using ComPASS³ demonstrates the ability of TopS to be implemented to gain new insights from independently generated protein interaction network datasets.

In Conclusion, we greatly appreciate both Reviewers’ insightful analysis and constructive comments regarding the TopS approach for analysing protein interaction networks. We have addressed all the comments and significantly revised the main body of the manuscript and added

substantial analysis and data to the supplementary materials. As a result of this process, we believe that the manuscript is dramatically improved.

References Used in Reviewer Response

1. Choi, H. *et al.*, SAINT: probabilistic scoring of affinity purification-mass spectrometry data. *Nat Methods* **8** (1), 70 (2011).
2. Sowa, M. E., Bennett, E. J., Gygi, S. P., and Harper, J. W., Defining the human deubiquitinating enzyme interaction landscape. *Cell* **138** (2), 389 (2009).
3. Hauri, S. *et al.*, A High-Density Map for Navigating the Human Polycomb Complexome. *Cell Rep* **17** (2), 583 (2016).
4. Hart, G. T., Lee, I., and Marcotte, E. R., A high-accuracy consensus map of yeast protein complexes reveals modular nature of gene essentiality. *BMC Bioinformatics* **8**, 236 (2007).
5. Isidor, B. *et al.*, De Novo Truncating Mutations in the Kinetochore-Microtubules Attachment Gene CHAMP1 Cause Syndromic Intellectual Disability. *Human mutation* **37** (4), 354 (2016).
6. Choi, H., Fermin, D., and Nesvizhskii, A. I., Significance analysis of spectral count data in label-free shotgun proteomics. *Mol Cell Proteomics* **7** (12), 2373 (2008).
7. Lum, P. Y. *et al.*, Extracting insights from the shape of complex data using topology. *Sci Rep* **3**, 1236 (2013).
8. Sardu, M. E., Gilmore, J. M., Groppe, B., Florens, L., and Washburn, M. P., Identification of Topological Network Modules in Perturbed Protein Interaction Networks. *Sci Rep* **7**, 43845 (2017).
9. Sardu, M. E. *et al.*, Conserved abundance and topological features in chromatin-remodeling protein interaction networks. *EMBO Rep* **16** (1), 116 (2015).
10. Mellacheruvu, D. *et al.*, The CRAPome: a contaminant repository for affinity purification-mass spectrometry data. *Nat Methods* **10** (8), 730 (2013).
11. Knight, J. D. R. *et al.*, ProHits-viz: a suite of web tools for visualizing interaction proteomics data. *Nat Methods* **14** (7), 645 (2017).
12. Huttlin, E. L. *et al.*, The BioPlex Network: A Systematic Exploration of the Human Interactome. *Cell* **162** (2), 425 (2015).
13. Gavin, A. C. *et al.*, Proteome survey reveals modularity of the yeast cell machinery. *Nature* **440** (7084), 631 (2006).

Reviewers' comments:

Reviewer #1 (Remarks to the Author):

The authors have now satisfactorily addressed the questions and comments in the review, and my recommendation is now to proceed with publication.

Reviewer #3 (Remarks to the Author):

I was not a reviewer of the first version, and was asked to comment on the response of the authors to the original review, and the suitability of the revised manuscript for publication. The manuscript by Sardu et al. complements existing software to score interactions by focusing on modularity in the interactions, and the authors suggest that the approach provides evidence for direct interactions in AP-MS data. The tool described, TopS, models the bait-prey interactions across a whole dataset, providing scores that highlight enriched interactions "in each bait relative to other baits in a larger biological context". Overall, I believe that the tool presented may be useful to the scientific community, though there are functionalities that do overlap with CompPASS or specificity scores as reported by others (e.g. ProHits-viz Prey Specificity generates similar plots to some of those presented in the current manuscript).

My overall assessment is that the authors have been largely responsive to the initial reviews, though I have specific comments that should be addressed.

Main comments:

1. It is really unclear how the "SAINT" analyses were performed: the authors referred to the CRAPome and FC-A/FC-B, two simple fold-change metrics distinct from SAINT, which suggests that what they are reporting are not SAINT scores at all. SAINT scores should be on a 0-1 scale, or represented by a Bayesian FDR score. Also note that SAINT scoring is affected by optional parameter selection and that such parameters must be specified alongside the version of SAINT used (original SAINT versus SAINTexpress in the CRAPome). Because of this issue, the comparison listed in the rebuttal as well as the associated Sup figure 10, table 10A needs to be revisited. In this respect, SAINT or SAINTexpress would be expected to behave in a manner similar to QSPEC to provide a list of high-confidence interactors by scoring against the controls, though it may be more stringent in scoring than QSPEC since it forces separation of the distributions between controls and test samples.
2. Given that TopS is used post QSPEC analysis (or at least this is what it seems to me based on Figure 1) and that there is a pre-filtering happening pre-TopS analysis, it seems disingenuous to refer to the comparative analyses as TopS versus CompPASS versus SAINT (again, as far as I can tell, this is NOT SAINT). I would recommend that 1) the authors refer to their analysis as QSPEC/TopS (not TopS alone); 2) that they perform a proper precision recall analysis, across their entire dataset and individual baits, using increments of the "n" highest scores. This should replace Supplementary Figure 10.
3. Presentation of violin plots vs score is also misleading since scaling is so different. While the authors use this as part of their arguments in favor of their score, this is a rather weak argument (it really matters very little what scale you select for scoring if the metrics are on spot) and I recommend that this part (line 336-340) be deleted, alongside the corresponding Sup Figure panel.
4. The identification of "direct" interactors from the data as high TopS scoring proteins is interesting, but in the absence of clear metrics associated with predictive power, I think this claim could be mis-interpreted by readers. The authors should revise the text to decrease the claims of direct/indirect discrimination or provide useful guidelines for their readers.
5. I cannot help but feeling that the manuscript is too long for what it describes, which may decrease its uptake by the community. I would recommend that the authors try making the description of the results in particular more concise.

6. The authors have not commented on the minimum number of

Minor comments:

1. Is the thresholding from the CRAPome performed directly in the user interface, or must it be done manually from the downloaded file (e.g. in Excel). I could not find clear instructions in the manuscript (personally, I also thought that the 2% filter is excessive, especially after QSPEC scoring) but since this is an optional filter, this is probably fine).

2. It is mentioned in the discussion that the method is compatible, e.g. with intensity data, but no examples are provided. It would have been nice for the authors to present at least a summary analysis.

Throughout: CompPASS (not ComPASS); CRAPome (not CrapOme); BioGRID (not BIOGRID)

The inset legends on some of the figures are difficult to read with the selected point size.

Line 74: perhaps add the mention of the polycomb complex?

Figure 7, panel A: some of the text seems covered by a white box

Line 223: eIF3 (not e1F)

Line 292: IES6?

Line 331: INO80

Line 342: AP-MS

Line 383: 3 datasets?

Line 383: IES6?

Response to the Reviewers' Comments

Reviewer #1 (Remarks to the Author):

The authors have now satisfactorily addressed the questions and comments in the review, and my recommendation is now to proceed with publication.

Response

We are grateful for Reviewer#1's excellent suggestions regarding our initial submission and we are pleased that we were able to address their suggestions regarding the first revision.

Reviewer #3 (Remarks to the Author):

I was not a reviewer of the first version, and was asked to comment on the response of the authors to the original review, and the suitability of the revised manuscript for publication. The manuscript by Sardu et al. complements existing software to score interactions by focusing on modularity in the interactions, and the authors suggest that the approach provides evidence for direct interactions in AP-MS data. The tool described, TopS, models the bait-prey interactions across a whole dataset, providing scores that highlight enriched interactions "in each bait relative to other baits in a larger biological context". Overall, I believe that the tool presented may be useful to the scientific community, though there are functionalities that do overlap with CompPASS or specificity scores as reported by others (e.g. ProHits-viz Prey Specificity generates similar plots to some of those presented in the current manuscript).

My overall assessment is that the authors have been largely responsive to the initial reviews, though I have specific comments that should be addressed.

Response

We would like to thank Reviewer #3 for their support and positive evaluation of our responsiveness to Reviewer #2's comments regarding our initial submission. Below, we address the additional comments provided by Reviewer #3 and the associated changes we made in the manuscript in response to Reviewer #3's concerns.

Main comments:

1. It is really unclear how the "SAINT" analyses were performed: the authors referred to the CRAPome and FC-A/FC-B, two simple fold-change metrics distinct from SAINT, which suggests that what they are reporting are not SAINT scores at all. SAINT scores should be on a 0-1 scale, or represented by a Bayesian FDR score. Also note that SAINT scoring is affected by optional parameter selection and that such parameters must be specified alongside the version of SAINT used (original SAINT versus SAINTexpress in the CRAPome). Because of this issue, the comparison listed in the rebuttal as well as the associated Sup figure 10, table 10A needs to be revisited. In this respect, SAINT or SAINTexpress would be expected to behave in a manner similar to QSPEC to provide a list of high-confidence interactors by scoring against the controls, though it may be more stringent in scoring than QSPEC since it forces separation of the distributions between controls and test samples.

Response

This was a valuable recommendation and that we have addressed in the revised manuscript. We indeed previously focused only on the FC(Empirical) score and secondary scores from SAINT 2.0, which is available to use within the Contaminant Repository for Affinity Purification (CRAPome.org)¹. Following the reviewer' comments we have now added the SAINT scores and Bayesian false discovery rate-BFDR for the three datasets.

We performed 41 SAINTexpress (SAINTexpress-v3.6.3-2018) analyses and the results are now in the Supplementary tables 8, 9, 10 and 11. We downloaded the SAINTexpress-v3.6.3-2018 from sourceforge at <https://sourceforge.net/projects/saint-apms/files/> and ran it with the default parameters. For every bait in our datasets we constructed three files: bait file, prey file and interaction file. These SAINT score results were also verified using Galaxy platform at <http://apostl.moffitt.org/> using the same files as for the SAINTexpress linux version.

We have now added the details of performing SAINTexpress to the manuscript in the Supplementary File on page 2. In addition, several supplementary tables have been updated. Supplementary Table 8 (worksheet2: SAINTS_results) consists of SAINT scores and BFDR for the 17 baits. Supplementary Table 9 (worksheet2: SAINT_results and worksheet5: Croslinking Comparison) consists of SAINT scores and BDFR for the yeast INO80 14 baits and TopS, QSPEC, CompPASS and SAINT scores and BFDR for the known interactions from crosslinking data. Supplementary Table 10 (worksheet2: SAINT_results and worksheet5: Croslinking Comparison) consists of SAINT scores and BDFR for the yeast SWI/SNF 10 baits and TopS, QSPEC, CompPASS and SAINT scores and BFDR for the known interactions from crosslinking data. Lastly, Supplementary Table 11 (worksheet4: intersection_biogrid) consists of TopS, QSPEC, CompPASS and SAINT scores and BFDR for the interactions detected in BIOGRID. The referee is right that SAINT is more stringent in scoring than QSPEC.

2. Given that TopS is used post QSPEC analysis (or at least this is what it seems to me based on Figure 1) and that there is a pre-filtering happening pre-TopS analysis, it seems disingenuous to refer to the comparative analyses as TopS versus CompPASS versus SAINT (again, as far as I can tell, this is NOT SAINT). I would recommend that 1) the authors refer to their analysis as QSPEC/TopS (not TopS alone);

Response

It is important to note that Reviewers #1 and #2 from the original submission requested the comparison of TopS versus existing pipelines, and we used CompPASS² and SAINT³ in this analysis for the first revision. TopS is complimentary to CompPASS² and SAINT³ and can be used in addition to these pipelines. We stress this in several locations in the manuscript. TopS in principle could be used with other pipelines like MUSE⁴ and MIST⁵, but this was not tested here. TopS is likely best used in addition to existing pipelines and can likely be used in many ways, and therefore it is a standalone tool and should be named accordingly.

2) that they perform a proper precision recall analysis, across their entire dataset and individual baits, using increments of the "n" highest scores. This should replace Supplementary Figure 10.

Response

The main drawback of this approach is that it depends on the presence of ‘true’ positive and negative bait-prey interactions in the MS data set. The panel presented in supplementary figure 10 was based on the excellent suggestion of Reviewer #2 from the original submission where we evaluated the accuracy of the score by using the recall for the set of the well-characterized interactions as described in the MIST⁵ publication from the Krogan Lab.

3. Presentation of violin plots vs score is also misleading since scaling is so different. While the authors use this as part of their arguments in favor of their score, this is a rather weak argument (it really matters very little what scale you select for scoring if the metrics are on spot) and I recommend that this part (line 336-340) be deleted, alongside the corresponding Sup Figure panel.

Response

As requested by Reviewer #3, we have removed Supplementary Figure 10 panels B-G and the discussion of this on page 16 of the main manuscript and in the supplementary information file. However, it is important to note that these figure panels and comparisons were included in the first revision of the manuscript in response to comments made by Reviewer #2 during their evaluation of the original submission.

4. The identification of “direct” interactors from the data as high TopS scoring proteins is interesting, but in the absence of clear metrics associated with predictive power, I think this claim could be mis-interpreted by readers. The authors should revise the text to decrease the claims of direct/indirect discrimination or provide useful guidelines for their readers.

Response

We have revised our usage of the term ‘direct’ in the manuscript by adding ‘potential’, or ‘correlated with’ for example, to soften the language and decrease the claims. In cases where no additional biochemical evidence exists, it is best to use tempered language. However, it is important to note that additional biochemical evidence for direct interactions does exist for several extreme values in the manuscript. For example, as described in the main body of the manuscript, in the XRCC5 affinity purification, XRCC6 is the highest scoring prey protein, which is the highest TopS value in the entire dataset, and in the XRCC6 affinity purification XRCC5 is the highest scoring prey protein. XRCC5 and XRCC6 are a known heterodimer⁶. There are other examples in the manuscript similar to this, like the cross linking data available for the yeast chromatin remodeling complexes described in the manuscript.

5. I cannot help but feeling that the manuscript is too long for what it describes, which may decrease its uptake by the community. I would recommend that the authors try making the description of the results in particular more concise.

Response

The revised manuscript that Reviewer #3 reviewed was a major revision from the first version of the manuscript in response to Reviewer #1’s and Reviewer #2’s comments. Also, while Reviewer #3 felt the manuscript was too long, Reviewer #1 was fully satisfied with the

revised manuscript. The major revisions that occurred based on Reviewer #1 and Reviewer #2's comments were the addition of the comparison of TopS to CompPASS and SAINT, and the reanalysis of the human polycomb dataset from Hauri *et. al*⁷ using TopS. These were major and important additions to the original manuscript.

Minor comments:

1. Is the thresholding from the CRAPome performed directly in the user interface, or must it be done manually from the downloaded file (e.g. in Excel). I could not find clear instructions in the manuscript (personally, I also thought that the 2% filter is excessive, especially after QSPEC scoring) but since this is an optional filter, this is probably fine).

Response

We have now included the details of how we performed CRAPome¹ in the manuscript on page 9. As we mentioned before, we selected this threshold of proteins present in 2% of the negative controls of the CRAPome¹ as a method to reduce the dataset to in order to highlight proteins enriched in the human DNA repair dataset. This threshold was used only to reduce the network size for Fig. 3B, it does not change any of our results or conclusions, and it is indeed an optional filter that is user dependent.

2. It is mentioned in the discussion that the method is compatible, e.g. with intensity data, but no examples are provided. It would have been nice for the authors to present at least a summary analysis.

Response

Since there is no data to support this statement provided in the manuscript, we have removed this statement. Providing such data and subsequent analysis is beyond the scope of the current manuscript.

*Throughout: CompPASS (not ComPASS); CRAPome (not CrapOme); BioGRID (not BIOGRID)
The inset legends on some of the figures are difficult to read with the selected point size.*

Line 74: perhaps add the mention of the polycomb complex?

Figure 7, panel A: some of the text seems covered by a white box

Line 223: eIF3 (not e1F)

Line 292: IES6?

Line 331: INO80

Line 342: AP-MS

Line 383: 3 datasets?

Line 383: IES6?

Response

We appreciate Reviewer #3's careful reading of our manuscript and we have made these important corrections.

References Cited in the Response Letter

1. Mellacheruvu, D. *et al.*, The CRAPome: a contaminant repository for affinity purification-mass spectrometry data. *Nat Methods* **10** (8), 730 (2013).
2. Sowa, M. E., Bennett, E. J., Gygi, S. P., and Harper, J. W., Defining the human deubiquitinating enzyme interaction landscape. *Cell* **138** (2), 389 (2009).
3. Choi, H. *et al.*, SAINT: probabilistic scoring of affinity purification-mass spectrometry data. *Nat Methods* **8** (1), 70 (2011).
4. Li, X. *et al.*, Defining the Protein-Protein Interaction Network of the Human Protein Tyrosine Phosphatase Family. *Mol Cell Proteomics* **15** (9), 3030 (2016).
5. Verschueren, E. *et al.*, Scoring Large-Scale Affinity Purification Mass Spectrometry Datasets with MiST. *Current protocols in bioinformatics* **49**, 8 19 1 (2015).
6. Rivera-Calzada, A., Spagnolo, L., Pearl, L. H., and Llorca, O., Structural model of full-length human Ku70-Ku80 heterodimer and its recognition of DNA and DNA-PKcs. *EMBO Rep* **8** (1), 56 (2007).
7. Hauri, S. *et al.*, A High-Density Map for Navigating the Human Polycomb Complexome. *Cell Rep* **17** (2), 583 (2016).

Reviewers' comments:

Reviewer #2 (Remarks to the Author):

I have looked at the authors' responses to Reviewer #3. I do not understand what the authors did to compare with the two other pipelines (SAINT and CompPASS). They write "We performed 41 SAINTexpress (SAINTexpress-v3.6.3-2018) analyses ...". There should be just one analysis per dataset, using all baits vs. the negative controls. This may be OK for CompPASS, but SAINT if for comparison against negative controls, not against other baits. Perhaps this explains the strange results shown in the Supplementary Figure 10. The authors should fix that.

I also agree with the comment that the manuscript feels too long because a lot of similar analyses/heatmaps/networks for these datasets were previously published by the same authors (including the Topological Network Analysis).

Response to the Reviewer's Comments

Reviewer #2 (Remarks to the Author):

I have looked at the authors' responses to Reviewer #3. I do not understand what the authors did to compare with the two other pipelines (SAINT and CompPASS). They write "We performed 41 SAINTexpress (SAINTexpress-v3.6.3-2018) analyses ...". There should be just one analysis per dataset, using all baits vs. the negative controls. This may be OK for CompPASS, but SAINT if for comparison against negative controls, not against other baits. Perhaps this explains the strange results shown in the Supplementary Figure 10. The authors should fix that.

Response

The reviewer is correct that the standard approach in the field is to compare bait affinity purification datasets with negative control datasets to determine protein complex content. This is what SAINT for example was designed to do. In this minor revision we show that CompPASS, SAINT, and TopS are all fully capable of determining protein complex components when comparing bait purifications to negative controls.

The differentiating aspect of TopS is that while it can carry out this type of comparison, it is particularly powerful when comparing bait purifications to bait purifications to find extreme values. In order to properly compare CompPASS, SAINT, and TopS we needed to implement all three in such a manner where we are comparing bait purifications to other bait purifications. For SAINT, this required the uncharacteristic and not previously implemented approach, as pointed out by Reviewer #2.

To clarify this point, we have made three minor revisions to the manuscript.

1). On page 15 of the main body of the manuscript in the first paragraph on the 'Comparison of TopS to Alternative Analysis Pipelines' We have revised the middle of the paragraph which now reads:

"First, as expected, we demonstrated that all three approaches effectively determine the components of protein complexes when bait purifications are compared to negative controls (Supplementary Fig. 10A). Next, we analyzed the recall of potential direct protein interactions from all three approaches using human DNA repair proteins from BioGRID³⁷ and from the crosslinking-based results of the INO80 and SWI/SNF dataset. In this case, every affinity purification was compared to all other affinity purifications in the dataset, which were considered positive controls (Supplementary File)."

2.) We added two sentences to the beginning of the Results section in the Supplementary File on page 3 which now reads:

"The standard approach in the field of analysis of protein complexes is to compare bait affinity purifications to negative control purifications. CompPASS¹, SAINT³, and TopS all are capable of fully determining components of protein complexes when bait affinity purifications

are compared to negative control datasets (Supplemental Figure 10A). All three methods performed well by recovering all 15 subunits of the INO80 and 12 subunits of SWI/SNF with high probability by SAINT or high scores by CompPASS and TopS (Supplementary Fig. 10A).”

3.) Lastly, we have revised Supplementary Figure 10 and its figure legend to include a panel (Supplementary Figure 10A) where the complete recall of protein complex components was achieved by CompPASS, SAINT, and TopS when comparing bait purifications to negative controls.